# How DNNs break the Curse of Dimensionality: Compositionality and Symmetry Learning

## Abstract

We show that deep neural networks (DNNs) can efficiently learn any composition of functions with bounded $F_1$-norm, which allows DNNs to break the curse of dimensionality in ways that shallow networks cannot. More specifically, we derive a generalization bound that combines a covering number argument for compositionality, and the $F_1$-norm (or the related Barron norm) for large width adaptivity. We show that the global minimizer of the regularized loss of DNNs can fit for example the composition of two functions $f^* = h \circ g$ from a small number of observations, assuming $g$ is smooth/regular and reduces the dimensionality (e.g. $g$ could be the modulo map of the symmetries of $f^*$), so that $h$ can be learned in spite of its low regularity. The measures of regularity we consider is the Sobolev norm with different levels of differentiability, which is well adapted to the $F_1$ norm. We compute scaling laws empirically and observe phase transitions depending on whether $g$ or $h$ is harder to learn, as predicted by our theory.

## 1 Introduction

One of the fundamental features of DNNs is their ability to generalize even when the number of neurons (and of parameters) is so large that the network could fit almost any function [46]. Actually DNNs have been observed to generalize best when the number of neurons is infinite [8, 21, 20]. The now quite generally accepted explanation to this phenomenon is that DNNs have an implicit bias coming from the training dynamic where properties of the training algorithm lead to networks that generalize well. This implicit bias is quite well understood in shallow networks [11, 36], in linear networks [24, 30], or in the NTK regime [28], but it remains ill-understood in the general deep nonlinear case.

In both shallow networks and linear networks, one observes a bias towards small parameter norm (either implicit [12] or explicit in the presence of weight decay [42]). Thanks to tools such as the $F_1$-norm [5], or the related Barron norm [44], or more generally the representation cost [14], it is possible to describe the family of functions that can be represented by shallow networks or linear networks with a finite parameter norm. This was then leveraged to prove uniform generalization bounds (based on Rademacher complexity) over these sets [5], which depend only on the parameter norm, but not on the number of neurons or parameters.

Similar bounds have been proposed for DNNs [7, 6, 39, 33, 25, 40], relying on different types of norms on the parameters of the network. But it seems pretty clear that we have not yet identified the 'right' complexity measure for deep networks, as there remains many issues: these bounds are typically orders of magnitude too large [29, 23], and they tend to explode as the depth $L$ grows [40].

Two families of bounds are particularly relevant to our analysis: bounds based on covering numbers which rely on the fact that one can obtain a covering of the composition of two function classes from

covering of the individual classes [7, 25], and path-norm bounds which extend the techniques behind the $F_1$-norm bound from shallow networks to the deep case [32, 6, 23].

Another issue is the lack of approximation results to accompany these generalization bounds: many different complexity measures $R(\theta)$ on the parameters $\theta$ of DNNs have been proposed along with guarantees that the generalization gap will be small as long as $R(\theta)$ is bounded, but there are often little to no result describing families of functions that can be approximated with a bounded $R(\theta)$ norm. The situation is much clearer in shallow networks, where we know that certain Sobolev spaces can be approximated with bounded $F_1$-norm [5].

We will focus on approximating composition of Sobolev functions, and obtaining close to optimal rates. This is quite similar to the family of tasks considered [39], though the complexity measure we consider is quite different, and does not require sparsity of the parameters.

## 1.1 Contribution

We consider Accordion Networks (AccNets), which are the composition of multiple shallow networks $f_{L:1} = f_L \circ \cdots \circ f_1$, we prove a uniform generalization bound $\mathcal{L}(f_{L:1}) - \tilde{\mathcal{L}}_N(f_{L:1}) \lesssim R(f_1, \ldots, f_L) \frac{\log N}{\sqrt{N}}$, for a complexity measure

$$R(f_1, \ldots, f_L) = \prod_{\ell=1}^{L} Lip(f_\ell) \sum_{\ell=1}^{L} \frac{\|f_\ell\|_{F_1}}{Lip(f_\ell)} \sqrt{d_\ell + d_{\ell-1}}$$

that depends on the $F_1$-norms $\|f_\ell\|_{F_1}$ and Lipschitz constant $Lip(f_\ell)$ of the subnetworks, and the intermediate dimensions $d_0, \ldots, d_L$. This use of the $F_1$-norms makes this bound independent of the widths $w_1, \ldots, w_L$ of the subnetworks, though it does depend on the depth $L$ (it typically grows linearly in $L$ which is still better than the exponential growth often observed).

Any traditional DNN can be mapped to an AccNet (and vice versa), by splitting the middle weight matrices $W_\ell$ with SVD $USV^T$ into two matrices $U\sqrt{S}$ and $\sqrt{S}V^T$ to obtain an AccNet with dimensions $d_\ell = \mathrm{Rank}W_\ell$, so that the bound can be applied to traditional DNNs with bounded rank.

We then show an approximation result: any composition of Sobolev functions $f^* = f_{L^*}^* \circ \cdots \circ f_1^*$ can be approximated with a network with either a bounded complexity $R(\theta)$ or a slowly growing one. Thus under certain assumptions one can show that DNNs can learn general compositions of Sobolev functions. This ability can be interpreted as DNNs being able to learn symmetries, allowing them to avoid the curse of dimensionality in settings where kernel methods or even shallow networks suffer heavily from it.

Empirically, we observe a good match between the scaling laws of learning and our theory, as well as qualitative features such as transitions between regimes depending on whether it is harder to learn the symmetries of a task, or to learn the task given its symmetries.

## 2 Accordion Neural Networks and ResNets

Our analysis is most natural for a slight variation on the traditional fully-connected neural networks (FCNNs), which we call Accordion Networks, which we define here. Nevertheless, all of our results can easily be adapted to FCNNs.

Accordion Networks (AccNets) are simply the composition of $L$ shallow networks, that is $f_{L:1} = f_L \circ \cdots \circ f_1$ where $f_\ell(z) = W_\ell \sigma(V_\ell z + b_\ell)$ for the nonlinearity $\sigma : \mathbb{R} \to \mathbb{R}$, the $d_\ell \times w_\ell$ matrix $W_\ell$, $w_\ell \times d_{\ell-1}$ matrix $V_\ell$, and $w_\ell$-dim. vector $b_\ell$, and for the widths $w_1, \ldots, w_L$ and dimensions $d_0, \ldots, d_L$. We will focus on the ReLU $\sigma(x) = \max\{0, x\}$ for the nonlinearity. The parameters $\theta$ are made up of the concatenation of all $(W_\ell, V_\ell, b_\ell)$. More generally, we denote $f_{\ell_2:\ell_1} = f_{\ell_2} \circ \cdots \circ f_{\ell_1}$ for any $1 \leq \ell_1 \leq \ell_2 \leq L$.

We will typically be interested in settings where the widths $w_\ell$ is large (or even infinitely large), while the dimensions $d_\ell$ remain finite or much smaller in comparison, hence the name accordion.

If we add residual connections, i.e. $f_{1:L}^{res} = (f_L + id) \circ \cdots \circ (f_1 + id)$ for the same shallow nets $f_1, \ldots, f_L$ we recover the typical ResNets.

81 *Remark.* The only difference between AccNets and FCNNs is that each weight matrix $M_\ell$ of the
82 FCNN is replaced by a product of two matrices $M_\ell = V_\ell W_{\ell-1}$ in the middle of the network (such a
83 structure has already been proposed [34]). Given an AccNet one can recover an equivalent FCNN by
84 choosing $M_\ell = V_\ell W_{\ell-1}$, $M_0 = V_0$ and $M_{L+1} = W_L$. In the other direction there could be multiple
85 ways to split $M_\ell$ into the product of two matrices, but we will focus on taking $V_\ell = U\sqrt{S}$ and
86 $W_{\ell-1} = \sqrt{S}V^T$ for the SVD decomposition $M_\ell = USV^T$, along with the choice $d_\ell = \mathrm{Rank}M_\ell$.

## 2.1 Learning Setup

88 We consider a traditional learning setup, where we want to find a function $f : \Omega \subset \mathbb{R}^{d_{in}} \to \mathbb{R}^{d_{out}}$
89 that minimizes the population loss $\mathcal{L}(f) = \mathbb{E}_{x \sim \pi}[\ell(x, f(x))]$ for an input distribution $\pi$ and a
90 $\rho$-Lipschitz and $\rho$-bounded loss function $\ell(x, y) \in [0, B]$. Given a training set $x_1, \ldots, x_N$ of size $N$
91 we approximate the population loss by the empirical loss $\tilde{\mathcal{L}}_N(f) = \frac{1}{N}\sum_{i=1}^N \ell(x_i, f(x_i))$ that can be
92 minimized.

93 To ensure that the empirical loss remains representative of the population loss, we will prove high
94 probability bounds on the generalization gap $\tilde{\mathcal{L}}_N(f) - \mathcal{L}(f)$ uniformly over certain functions families
95 $f \in \mathcal{F}$.

96 For **regression tasks**, we assume the existence of a true function $f^*$ and try to minimize the distance
97 $\ell(x, y) = \|f^*(x) - y\|^p$ for $p \geq 1$. If we assume that $f^*(x)$ and $y$ are uniformly bounded then one
98 can easily show that $\ell(x, y)$ is bounded and Lipschitz. We are particularly interested in the cases
99 $p \in \{1, 2\}$, with $p = 2$ representing the classical MSE, and $p = 1$ representing a $L_1$ distance. The
100 $p = 2$ case is amenable to 'fast rates' which take advantage of the fact that the loss increases very
101 slowly around the optimal solution $f^*$, We do not prove such fast rates (even though it might be
102 possible) so we focus on the $p = 1$ case.

103 For **classification tasks** on $k$ classes, we assume the existence of a 'true class' function $f^* : \Omega \to$
104 $\{1, \ldots, k\}$ and want to learn a function $f : \Omega \to \mathbb{R}^k$ such that the largest entry of $f(x)$ is the $f^*(k)$-th
105 entry. One can consider the hinge cost $\ell(x, y) = \max\{0, 1 - (y_{f^*(k)} - \max_{i \neq f^*(x)} y_i)\}$, which is
106 zero whenever the margin $y_{f^*(k)} - \max_{i \neq f^*(x)} y_i$ is larger than 1 and otherwise equals 1 minus the
107 margin. The hinge loss is Lipschitz and bounded if we assume bounded outputs $y = f(x)$. The
108 cross-entropy loss also fits our setup.

## 3 Generalization Bound for DNNs

110 The reason we focus on accordion networks is that there exists generalization bounds for shallow
111 networks [5, 44], that are (to our knowledge) widely considered to be tight, which is in contrast to the
112 deep case, where many bounds exist but no clear optimal bound has been identified. Our strategy
113 is to extend the results for shallow nets to the composition of multiple shallow nets, i.e. AccNets.
114 Roughly speaking, we will show that the complexity of an AccNet $f_\theta$ is bounded by the sum of the
115 complexities of the shallow nets $f_1, \ldots, f_L$ it is made of.

116 We will therefore first review (and slightly adapt) the existing generalization bounds for shallow
117 networks in terms of their so-called $F_1$-norm [5], and then prove a generalization bound for deep
118 AccNets.

### 3.1 Shallow Networks

120 The complexity of a shallow net $f(x) = W\sigma(Vx + b)$, with weights $W \in \mathbb{R}^{w \times d_{out}}$ and
121 $V \in \mathbb{R}^{d_{in} \times w}$, can be bounded in terms of the quantity $C = \sum_{i=1}^w \|W_{\cdot i}\|\sqrt{\|V_{i\cdot}\|^2 + b_i^2}$.
122 First note that the rescaled function $\frac{1}{C}f$ can be written as a convex combination $\frac{1}{C}f(x) =$
123 $\sum_{i=1}^w \frac{\|W_{\cdot i}\|\sqrt{\|V_{i\cdot}\|^2 + b_i^2}}{C}\bar{W}_{\cdot i}\sigma(\bar{V}_{i\cdot}x + \bar{b}_i)$ for $\bar{W}_{\cdot i} = \frac{W_{\cdot i}}{\|W_{\cdot i}\|}$, $\bar{V}_{i\cdot} = \frac{V_{i\cdot}}{\sqrt{\|V_{i\cdot}\|^2 + b_i^2}}$, and $\bar{b}_i = \frac{b_i}{\sqrt{\|V_{i\cdot}\|^2 + b_i^2}}$,
124 since the coefficients $\frac{\|W_{\cdot i}\|\sqrt{\|V_{i\cdot}\|^2 + b_i^2}}{C}$ are positive and sum up to 1. Thus $f$ belongs to $C$ times the
125 convex hull

$$B_{F_1} = \mathrm{Conv}\left\{x \mapsto w\sigma(v^T x + b) : \|w\|^2 = \|v\|^2 + b^2 = 1\right\}.$$

126  We call this the $F_1$-ball since it can be thought of as the unit ball w.r.t. the $F_1$-norm $\|f\|_{F_1}$ which we
127  define as the smallest positive scalar $s$ such that[1] $\frac{1}{s} f \in B_{F_1}$. For more details in the single output
128  case, see [5].

129  The generalization gap over any $F_1$-ball can be uniformly bounded with high probability:

130  **Theorem 1.** *For any input distribution $\pi$ supported on the $L_2$ ball $B(0, b)$ with radius $b$, we have*
131  *with probability $1 - \delta$, over the training samples $x_1, \ldots, x_N$, that for all $f \in B_{F_1}(0, R) = R \cdot B_{F_1}$*

$$\mathcal{L}(f) - \tilde{\mathcal{L}}_N(f) \le \rho b R \sqrt{d_{in} + d_{out}} \frac{\log N}{\sqrt{N}} + c_0 \sqrt{\frac{2 \log 2/\delta}{N}}$$

132  This theorem is a slight variation of the one found in [5]: we simply generalize it to multiple outputs,
133  and also prove it using a covering number argument instead of a direct computation of the Rademacher
134  complexity, which will be key to obtaining a generalization bound for the deep case. But due to this
135  change of strategy we pay a $\log N$ cost here and in our later results. We know that the $\log N$ term
136  can be removed in Theorem 1 by switching to a Rademacher argument, but we do not know whether
137  it can be removed in deep nets.

138  Notice how this bound does not depend on the width $w$, because the $F_1$-norm (and the $F_1$-ball)
139  themselves do not depend on the width. This matches with empirical evidence that shows that
140  increasing the width does not hurt generalization [8, 21, 20].

141  To use Theorem 1 effectively we need to be able to guarantee that the learned function will have a
142  small enough $F_1$-norm. The $F_1$-norm is hard to compute exactly, but it is bounded by the parameter
143  norm: if $f(x) = W\sigma(Vx + b)$, then $\|f\|_{F_1} \le \frac{1}{2} \left( \|W\|_F^2 + \|V\|_F^2 + \|b\|^2 \right)$, and this bound is tight
144  if the width $w$ is large enough and the parameters are chosen optimally. Adding weight decay/$L_2$-
145  regularization to the cost then leads to bias towards learning with small $F_1$ norm.

## 3.2 Deep Networks

147  Since an AccNet is simply the composition of multiple shallow nets, the functions represented by an
148  AccNet is included in the set of composition of $F_1$ balls. More precisely, if $\|W_\ell\|^2 + \|V_\ell\|^2 + \|b_\ell\|^2 \le$
149  $2R_\ell$ then $f_{L:1}$ belongs to the set $\{g_L \circ \cdots \circ g_1 : g_\ell \in B_{F_1}(0, R_\ell)\}$ for some $R_\ell$, which is width
150  agnostic.

151  As already noticed in [7], the covering number number is well-behaved under composition, this
152  allows us to bound the complexity of AccNets in terms of the individual shallow nets it is made up of:

153  **Theorem 2.** *Consider an accordion net of depth $L$ and widths $d_L, \ldots, d_0$, with corresponding set of*
154  *functions $\mathcal{F} = \{f_{L:1} : \|f_\ell\|_{F_1} \le R_\ell, Lip(f_\ell) \le \rho_\ell\}$. With probability $1 - \delta$ over the sampling of the*
155  *training set $X$ from the distribution $\pi$ supported in $B(0, b)$, we have for all $f \in \mathcal{F}$*

$$\mathcal{L}(f) - \tilde{\mathcal{L}}_N(f) \le C \rho b \rho_{L:1} \sum_{\ell=1}^{L} \frac{R_\ell}{\rho_\ell} \sqrt{d_\ell + d_{\ell-1}} \frac{\log N}{\sqrt{N}} (1 + o(1)) + c_0 \sqrt{\frac{2 \log 2/\delta}{N}}.$$

156  Theorem 2 can be extended to ResNets $(f_L + id) \circ \cdots \circ (f_1 + id)$ by simply replacing the Lipschitz
157  constant $Lip(f_\ell)$ by $Lip(f_\ell + id)$.

158  The Lipschitz constants $Lip(f_\ell)$ are difficult to compute exactly, so it is easiest to simply bound it
159  by the product of the operator norms $Lip(f_\ell) \le \|W_\ell\|_{op} \|V_\ell\|_{op}$, but this bound can often be quite
160  loose. The fact that our bound depends on the Lipschitz constants rather than the operator norms
161  $\|W_\ell\|_{op}, \|V_\ell\|_{op}$ is thus a significant advantage.

162  This bound can be applied to a FCNNs with weight matrices $M_1, \ldots, M_{L+1}$, by replacing the middle
163  $M_\ell$ with SVD decomposition $USV^T$ in the middle by two matrices $W_{\ell-1} = \sqrt{S}V^T$ and $V_\ell = U\sqrt{S}$,
164  so that the dimensions can be chosen as the rank $d_\ell = \text{Rank} M_{\ell+1}$. The Frobenius norm of the new
165  matrices equal the nuclear norm of the original one $\|W_{\ell-1}\|_F^2 = \|V_\ell\|_F^2 = \|M_\ell\|_*$. Some bounds

---

[1]This construction can be used for any convex set $B$ that is symmetric around zero ($B = -B$) to define a
norm whose unit ball is $B$. This correspondence between symmetric convex sets and norms is well known.

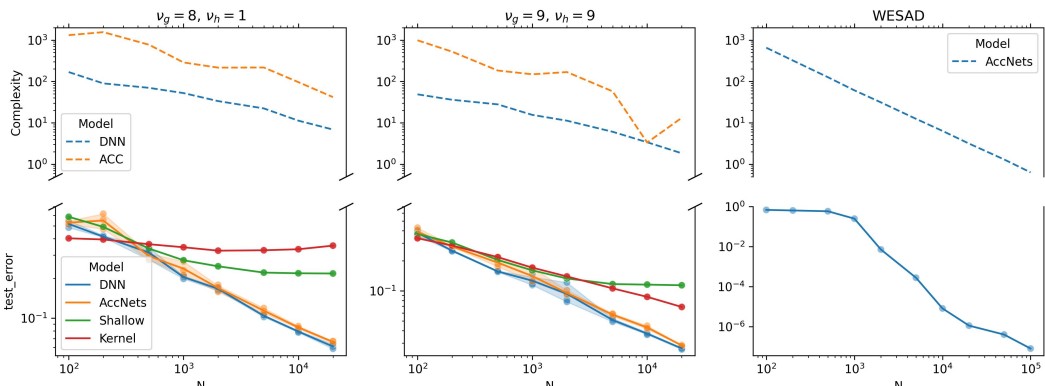

Figure 1: Visualization of scaling laws. We observe that deep networks (either AccNets or DNNs) achieve better scaling laws than kernel methods or shallow networks on certain compositional tasks, in agreement with our theory. We also see that our new generalization bounds approximately recover the right saling laws (even though they are orders of magnitude too large overall). We consider a compositional true function $f^* = h \circ g$ where $g$ maps from dimension 15 to 3 while h maps from 3 to 20, and we denote $\nu_g, \nu_h$ for the number of times $g, h$ are differentiable. In the first plot $\nu_g = 8, \nu_h = 1$ so that $g$ is easy to learn while $h$ is hard, whereas in the second plot $\nu_g = 9, \nu_h = 9$, so both $g$ and $h$ are relatively easier. The third plot presents the decay in test error and generalization bounds for networks evaluated using the real-world dataset, WESAD [37].

assuming rank sparsity of the weight matrices also appear in [41]. And several recent results have shown that weight-decay leads to a low-rank bias on the weight matrices of the network [27, 26, 19] and replacing the Frobenius norm regularization with a nuclear norm regularization (according to the above mentioned equivalence) will only increase this low-rank bias.

We compute in Figure 1 the upper bound of Theorem 2 for both AccNets and DNNs, and even though we observe a very large gap (roughly of order $10^3$), we do observe that it captures rate/scaling of the test error (the log-log slope) well. So this generalization bound could be well adapted to predicting rates, which is what we will do in the next section.

*Remark.* Note that if one wants to compute this upper bound in practical setting, it is important to train with $L_2$ regularization until the parameter norm also converges (this often happens after the train and test loss have converged). The intuition is that at initialization, the weights are initialized randomly, and they contribute a lot to the parameter norm, and thus lead to a larger generalization bound. During training with weight decay, these random initial weights slowly vanish, thus leading to a smaller parameter norm and better generalization bound. It might be possible to improve our generalization bounds to take into account the randomness at initialization to obtain better bounds throughout training, but we leave this to future work.

## 4 Breaking the Curse of Dimensionality with Compositionality

In this section we study a large family of functions spaces, obtained by taking compositions of Sobolev balls. We focus on this family of tasks because they are well adapted to the complexity measure we have identified, and because kernel methods and even shallow networks do suffer from the curse of dimensionality on such tasks, whereas deep networks avoid it (e.g. Figure 1).

More precisely, we will show that these sets of functions can be approximated by a AccNets with bounded (or in some cases slowly growing) complexity measure

$$R(f_1, \ldots, f_L) = \prod_{\ell=1}^{L} Lip(f_\ell) \sum_{\ell=1}^{L} \frac{\|f_\ell\|_{F_1}}{Lip(f_\ell)} \sqrt{d_\ell + d_{\ell-1}}.$$

This will then allow us show that AccNets can (assuming global convergence) avoid the curse of dimensionality, even in settings that should suffer from the curse of dimensionality, when the input dimension is large and the function is not very smooth (only a few times differentiable).

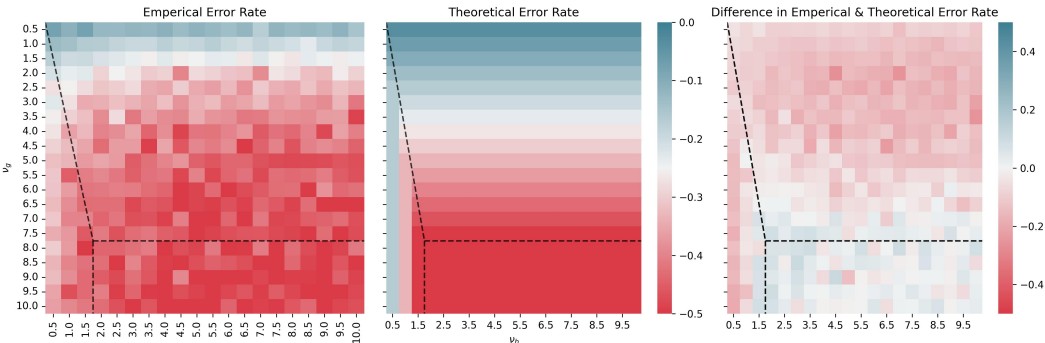

Figure 2: A comparison of empirical and theoretical error rates. The first plot illustrates the log decay rate of the test error with respect to the dataset size $N$ based on our empirical simulations. The second plot depicts the theoretical decay rate of the test error as discussed in Section 4.1, $-\min\{\frac{1}{2}, \frac{\nu_g}{d_{in}}, \frac{\nu_h}{d_{mid}}\}$. The final plot on the right displays the difference between the two. The lower left region represents the area where $g$ is easier to learn than $h$, the upper right where $h$ is easier to learn than $g$, and the lower right region where both $f$ and $g$ are easy.

.

## 4.1 Composition of Sobolev Balls

The family of Sobolev norms capture some notion of regularity of a function, as it measures the size of its derivatives. The Sobolev norm of a function $f : \mathbb{R}^{d_{in}} \to \mathbb{R}$ is defined in terms of its derivatives $\partial_x^\alpha f$ for some $d_{in}$-multi-index $\alpha$, namely the $W^{\nu,p}(\pi)$-Sobolev norm with integer $\nu$ and $p \geq 1$ is defined as

$$\|f\|_{W^{\nu,p}(\pi)}^p = \sum_{|\alpha| \leq \nu} \|\partial_x^\alpha f\|_{L_p(\pi)}^p .$$

Note that the derivative $\partial_x^\alpha f$ only needs to be defined in the 'weak' sense, which means that even non-differentiable functions such as the ReLU functions can actually have finite Sobolev norm.

The Sobolev balls $B_{W^{\nu,p}(\pi)}(0, R) = \{f : \|f\|_{W^{\nu,p}(\pi)} \leq R\}$ are a family of function spaces with a range of regularity (the larger $\nu$, the more regular). This regularity makes these spaces of functions learnable purely from the fact that they enforce the function $f$ to vary slowly as the input changes. Indeed we can prove the following generalization bound:

**Proposition 3.** *Given a distribution $\pi$ with support the $L_2$ ball with radius b, we have that with probability $1 - \delta$ for all functions $f \in \mathcal{F} = \{f : \|f\|_{W^{\nu,2}} \leq R, \|f\|_\infty \leq R\}$*

$$\mathcal{L}(f) - \tilde{\mathcal{L}}_N(f) \leq 2\rho C_1 R E_{\nu/d}(N) + c_0 \sqrt{\frac{2 \log 2/\delta}{N}}.$$

*where $E_r(N) = N^{-\frac{1}{2}}$ if $r > \frac{1}{2}$, $E_r(N) = N^{-\frac{1}{2}} \log N$ if $r = \frac{1}{2}$, and $E_r(N) = N^{-r}$ if $r < \frac{1}{2}$.*

But this result also illustrates the **curse of dimensionality:** the differentiability $\nu$ needs to scale with the input dimension $d_{in}$ to obtain a reasonable rate. If instead $\nu$ is constant and $d_{in}$ grows, then the number of datapoints $N$ needed to guarantee a generalization gap of at most $\epsilon$ scales exponentially in $d_{in}$, i.e. $N \sim \epsilon^{-\frac{d_{in}}{\nu}}$. One way to interpret this issue is that regularity becomes less and less useful the larger the dimension: knowing that similar inputs have similar outputs is useless in high dimension where the closest training point $x_i$ to a test point $x$ is typically very far away.

### 4.1.1 Breaking the Curse of Dimensionality with Compositionality

To break the curse of dimensionality, we need to assume some additional structure on the data or task which introduces an 'intrinsic dimension' that can be much lower than the input dimension $d_{in}$:

**Manifold hypothesis**: If the input distribution lies on a $d_{surf}$-dimensional manifold, the error rates typically depends on $d_{surf}$ instead of $d_{in}$ [38, 10].

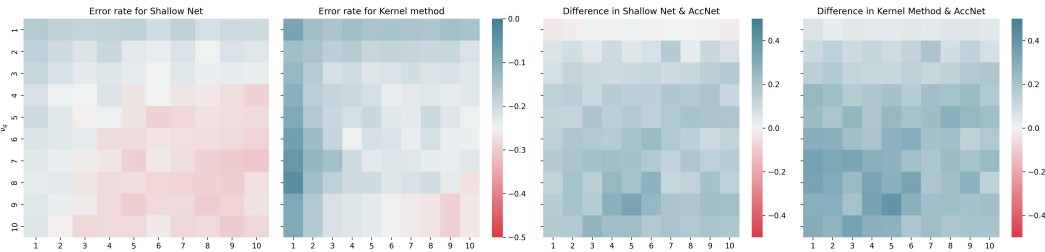

Figure 3: Comparing error rates for shallow and AccNets: shallow nets vs. AccNets, and kernel methods vs. AccNets. The left two graphs shows the empirical decay rate of test error with respect to dataset size (N) for both shallow nets and kernel methods. In contrast to our earlier empirical findings for AccNets, both shallow nets and kernel methods exhibit a slower decay rate in test error. The right two graphs present the differences in log decay rates between shallow nets and AccNets, as well as between kernel methods and AccNets. AccNets almost always obtain better rates, with a particularly large advantage at the bottom and middle-left.

.

**Known Symmetries:** If $f^*(g \cdot x) = f^*(x)$ for a group action $\cdot$ w.r.t. a group $G$, then $f^*$ can be written as the composition of a modulo map $g^* : \mathbb{R}^{d_{in}} \to \mathbb{R}^{d_{in}}/G$ which maps pairs of inputs which are equivalent up to symmetries to the same value (pairs $x, y$ s.t. $y = g \cdot x$ for some $g \in G$), and then a second function $h^* : \mathbb{R}^{d_{in}}/G \to \mathbb{R}^{d_{out}}$, then the complexity of the task will depend on the dimension of the modulo space $\mathbb{R}^{d_{in}}/G$ which can be much lower. If the symmetry is known, then one can for example fix $g^*$ and only learn $h^*$ (though other techniques exist, such as designing kernels or features that respect the same symmetries) [31].

**Symmetry Learning:** However if the symmetry is not known then both $g^*$ and $h^*$ have to be learned, and this is where we require feature learning and/or compositionality. Shallow networks are able to learn translation symmetries, since they can learn so-called low-index functions which satisfy $f^*(x) = f^*(Px)$ for some projection $P$ (with a statistical complexity that depends on the dimension of the space one projects into, not the full dimension [5, 2]). Low-index functions correspond exactly to the set of functions that are invariant under translation along the kernel $\ker P$. To learn general symmetries, one needs to learn both $h^*$ and the modulo map $g^*$ simultaneously, hence the importance of feature learning.

For $g^*$ to be learnable efficiently, it needs to be regular enough to not suffer from the curse of dimensionality, but many traditional symmetries actually have smooth modulo maps, for example the modulo map $g^*(x) = \|x\|^2$ for rotation invariance. This can be understood as a special case of composition of Sobolev functions, whose generalization gap can be bounded:

**Theorem 4.** *Consider the function set* $\mathcal{F} = \mathcal{F}_L \circ \cdots \circ \mathcal{F}_1$ *where* $\mathcal{F}_\ell = \left\{ f_\ell : \mathbb{R}^{d_{\ell-1}} \to \mathbb{R}^{d_\ell} \text{ s.t. } \|f_\ell\|_{W^{\nu_\ell,2}} \le R_\ell, \|f_\ell\|_\infty \le b_\ell, Lip(f_\ell) \le \rho_\ell \right\}$, *and let* $r_{min} = \min_\ell r_\ell$ *for* $r_\ell = \frac{\nu_\ell}{d_{\ell-1}}$, *then with probability* $1 - \delta$ *we have for all* $f \in \mathcal{F}$

$$\mathcal{L}(f) - \tilde{\mathcal{L}}_N(f) \le \rho C_0 \left( \sum_{\ell=1}^L (C_\ell \rho_{L:\ell+1} R_\ell)^{\frac{1}{r_{min}+1}} \right)^{r_{min}+1} E_{r_{min}}(N) + c_0 \sqrt{\frac{2 \log^2/\delta}{N}},$$

*where* $C_\ell$ *depends only on* $d_{\ell-1}, d_\ell, \nu_\ell, b_{\ell-1}$.

We see that only the smallest ratio $r_{min}$ matters when it comes to the rate of learning. And actually the above result could be slightly improved to show that the sum over all layers could be replaced by a sum over only the layers where the ratio $r_\ell$ leads to the worst rate $E_{r_\ell}(N) = E_{r_{min}}(N)$ (and the other layers contribute an asymptotically subdominant amount).

Coming back to the symmetry learning example, we see that the hardness of learning a function of the type $f^* = h \circ g$ with inner dimension $d_{mid}$ and regularities $\nu_g$ and $\nu_h$, the error rate will be (up to log terms) $N^{-\min\{\frac{1}{2}, \frac{\nu_g}{d_{in}}, \frac{\nu_h}{d_{mid}}\}}$. This suggests the existence of three regimes depending on which term attains the minimum: a regime where both $g$ and $h$ are easy to learn and we have $N^{-\frac{1}{2}}$ learning, a regime $g$ is hard, and a regime where $h$ is hard. The last two regimes differentiate between tasks

where learning the symmetry is hard and those where learning the function knowing its symmetries is hard.

In contrast, without taking advantage of the compositional structure, we expect $f^*$ to be only $\min\{\nu_g, \nu_h\}$ times differentiable, so trying to learn it as a single Sobolev function would lead to an error rate of $N^{-\min\{\frac{1}{2}, \frac{\min\{\nu_g, \nu_h\}}{d_{in}}\}} = N^{-\min\{\frac{1}{2}, \frac{\nu_g}{d_{in}}, \frac{\nu_h}{d_{in}}\}}$ which is no better than the compositional rate, and is strictly worse whenever $\nu_h < \nu_g$ and $\frac{\nu_h}{d_{in}} < \frac{1}{2}$ (we can always assume $d_{mid} \leq d_{in}$ since one could always choose $d = id$).

Furthermore, since multiple compositions are possible, one can imagine a hierarchy of symmetries that slowly reduce the dimensionality with less and less regular modulo maps. For example one could imagine a composition $f_L \circ \cdots \circ f_1$ with dimensions $d_\ell = d_0 2^{-\ell}$ and regularities $\nu_\ell = d_0 2^{-\ell}$ so that the ratios remain constant $r_\ell = \frac{d_0 2^{-\ell}}{d_0 2^{-\ell+1}} = \frac{1}{2}$, leading to an almost parametric rate of $N^{-\frac{1}{2}} \log N$ even though the function may only be $d_0 2^{-L}$ times differentiable. Without compositionality, the rate would only be $N^{-2^{-L}}$.

*Remark.* In the case of a single Sobolev function, one can show that the rate $E_{\nu/d}(N)$ is in some sense optimal, by giving an information theoretic lower bound with matching rate. A naive argument suggests that the rate of $E_{\min\{r_1, \ldots, r_L\}}(N)$ should similarly be optimal: assume that the minimum $r_\ell$ is attained at a layer $\ell$, then one can consider the subset of functions such that the image $f_{\ell-1:1}(B(0, r))$ contains a ball $B(z, r') \subset \mathbb{R}^{d_{\ell-1}}$ and that the function $f_{L:\ell+1}$ is $\beta$-non-contracting $\|f_{L:\ell+1}(x) - f_{L:\ell+1}(y)\| \geq \beta \|x - y\|$, then learning $f_{L:1}$ should be as hard as learning $f_\ell$ over the ball $B(z, r')$ (more rigorously this could be argued from the fact that any $\epsilon$-covering of $f_{L:1}$ can be mapped to an $\epsilon/\beta$-covering of $f_\ell$), thus forcing a rate of at least $E_{r_\ell}(N) = E_{\min\{r_1, \ldots, r_L\}}(N)$.

An analysis of minimax rates in a similar setting has been done in [22].

## 4.2 Breaking the Curse of Dimensionality with AccNets

Now that we know that composition of Sobolev functions can be easily learnable, even in settings where the curse of dimensionality should make it hard to learn them, we need to find a model that can achieve those rates. Though many models are possible [2], we focus on DNNs, in particular AccNets. Assuming convergence to a global minimum of the loss of sufficiently wide AccNets with two types of regularization, one can guarantee close to optimal rates:

**Theorem 5.** *Given a true function* $f^*_{L^*:1} = f^*_{L^*} \circ \cdots \circ f^*_1$ *going through the dimensions* $d^*_0, \ldots, d^*_{L^*}$, *along with a continuous input distribution* $\pi_0$ *supported in* $B(0, b_0)$, *such that the distributions* $\pi_\ell$ *of* $f^*_\ell(x)$ *(for* $x \sim \pi_0$*) are continuous too and supported inside* $B(0, b_\ell) \subset \mathbb{R}^{d^*_\ell}$. *Further assume that there are differentiabilities* $\nu_\ell$ *and radii* $R_\ell$ *such that* $\|f^*_\ell\|_{W^{\nu_\ell, 2}(B(0, b_\ell))} \leq R_\ell$, *and* $\rho_\ell$ *such that* $Lip(f^*_\ell) \leq \rho_\ell$. *For an infinite width AccNet with* $L \geq L^*$ *and dimensions* $d_\ell \geq d^*_1, \ldots, d^*_{L^*-1}$, *we have for the ratios* $\tilde{r}_\ell = \frac{\nu_\ell}{d^*_\ell + 3}$:

- *At a global minimizer* $\hat{f}_{L:1}$ *of the regularized loss* $f_1, \ldots, f_L \mapsto \tilde{\mathcal{L}}_N(f_{L:1}) + \lambda \prod_{\ell=1}^L Lip(f_\ell) \sum_{\ell=1}^L \frac{\|f_\ell\|_{F_1}}{Lip(f_\ell)} \sqrt{d_{\ell-1} + d_\ell}$, *we have* $\mathcal{L}(\hat{f}_{L:1}) = \tilde{O}(N^{-\min\{\frac{1}{2}, \tilde{r}_1, \ldots, \tilde{r}_{L^*}\}})$.

- *At a global minimizer* $\hat{f}_{L:1}$ *of the regularized loss* $f_1, \ldots, f_L \mapsto \tilde{\mathcal{L}}_N(f_{L:1}) + \lambda \prod_{\ell=1}^L \|f_\ell\|_{F_1}$, *we have* $\mathcal{L}(\hat{f}_{L:1}) = \tilde{O}(N^{-\frac{1}{2} + \sum_{\ell=1}^{L^*} \max\{0, \tilde{r}_\ell - \frac{1}{2}\}})$.

There are a number of limitations to this result. First we assume that one is able to recover the global minimizer of the regularized loss, which should be hard in general[3] (we already know from [5] that this is NP-hard for shallow networks and a simple $F_1$-regularization). Note that it is sufficient to recover a network $f_{L:1}$ whose regularized loss is within a constant of the global minimum, which

---

[2]One could argue that it would be more natural to consider compositions of kernel method models, for example a composition of random feature models. But this would lead to a very similar model: this would be equivalent to a AccNet where only the $W_\ell$ weights are learned, while the $V_\ell, b_\ell$ weights remain constant. Another family of models that should have similar properties is Deep Gaussian Processes [15].

[3]Note that the unregularized loss can be optimized polynomially, e.g. in the NTK regime [28, 3, 16], but this is an easier task than findinig the global minimum of the regularized loss where one needs to both fit the data, and do it with an minimal regularization term.

might be easier to guarantee, but should still be hard in general. The typical method of training with GD on the regularized loss is a greedy approach, which might fail in general but could recover almost optimal parameters under the right conditions (some results suggest that training relies on first order correlations to guide the network in the right direction [2, 1, 35]).

We propose two regularizations because they offer a tradeoff:

**First regularization:** The first regularization term leads to almost optimal rates, up to the change from $r_\ell = \frac{\nu_\ell}{d_\ell^*}$ to $r_\ell = \frac{\nu_\ell}{d_\ell^* + 3}$ which is negligible for large dimensions $d_\ell$ and differentiabilities $\nu_\ell$. The first problem is that it requires an infinite width at the moment, because we were not able to prove that a function with bounded $F_1$-norm and Lipschitz constant can be approximated by a sufficiently wide shallow networks with the same (or close) $F_1$-norm and Lipschitz constant (we know from [5] that it is possible without preserving the Lipschitzness). We are quite hopeful that this condition might be removed in future work.

The second and more significant problem is that the Lipschitz constants $Lip(f_\ell)$ are difficult to optimize over. For finite width networks it is in theory possible to take the max over all linear regions, but the complexity might be unreasonable. It might be more reasonable to leverage an implicit bias instead, such as a large learning rate, because a large Lipschitz constant implies that the nework is sensible to small changes in its parameters, so GD with a large learning rate should only converge to minima with a small Lipschitz constant (such a bias is described in [26]). It might also be possible to replace the Lipschitz constant in our generalization bounds, possibly along the lines of [43].

**Second regularization:** The second regularization term actually does not require an infinite width, only a sufficiently large one. Also its regularization term is equivalent to $\prod(\|W_\ell\|^2 + \|V_\ell\|^2 + \|b_\ell\|^2)$ which is much closer to the traditional $L_2$-regularization (and actually one could prove the same or very similar rates for $L_2$-regularization). The issue is that it lead to rates that could be far from optimal depending on the ratios $\tilde{r}_\ell$: it recovers the same rate as the first regularization term if no more than one ratio $\tilde{r}_\ell$ is smaller than $\frac{1}{2}$, but if many of these ratios are above $\frac{1}{2}$, it can be arbitrarily smaller.

In Figure 2, we compare the empirical rates (by doing a linear fit on a log-log plot of test error as a function of $N$) and the predicted optimal rates $\min\{\frac{1}{2}, \frac{\nu_g}{d_{in}}, \frac{\nu_h}{d_{mid}}\}$ and observe a pretty good match. Though surprisingly, it appears the the empirical rates tend to be slightly better than the theoretical ones.

*Remark.* As can be seen in the proof of Theorem 5, when the depth $L$ is strictly larger than the true depth $L^*$, one needs to add identity layers, leading to a so-called Bottleneck structure, which was proven to be optimal and observed empirically in [27, 26, 45]. These identity layers add a term that scales linearly in the additional depth $\frac{(L-L^*)d_{min}^*}{\sqrt{N}}$ to the first regularization, and an exponential prefactor $(2d_{min}^*)^{L-L^*}$ to the second. It might be possible to remove these factors by leveraging the bottleneck structure, or simply by switching to ResNets.

# 5 Conclusion

We have given a generalization bound for Accordion Networks and as an extension Fully-Connected networks. It depends on $F_1$-norms and Lipschitz constants of its shallow subnetworks. This allows us to prove under certain assumptions that AccNets can learn general compositions of Sobolev functions efficiently, making them able to break the curse of dimensionality in certain settings, such as in the presence of unknown symmetries.

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

# A   Experimental Setup[4]

In this section, we review our numerical experiments and their setup both on synthetic and real-world datasets in order to address theoretical results more clearly and intuitively.

## A.1   Dataset

### A.1.1   Emperical Dataset

The Matérn kernel is considered a generalization of the radial basis function (RBF) kernel. It controls the differentiability, or smoothness, of the kernel through the parameter $\nu$. As $\nu \to \infty$, the Matérn kernel converges to the RBF kernel, and as $\nu \to 0$, it converges to the Laplacian kernel, a 0-differentiable kernel. In this study, we utilized the Matérn kernel to generate Gaussian Process (GP) samples based on the composition of two Matérn kernels, $K_g$ and $K_h$, with varying differentiability in the range [0.5,10]×[0.5,10]. The input dimension ($d_{in}$) of the kernel, the bottleneck mid-dimension ($d_{mid}$), and the output dimension ($d_{out}$) are 15, 3, and 20, respectively.

This outlines the general procedure of our sampling method for synthetic data:

1. Sample the training dataset $X \in \mathbb{R}^{D \times d_{in}}$

2. From X, compute the $D \times D$ kernel $K_g$ with given $\nu_g$

3. From $K_g$, sample $Z \in \mathbb{R}^{D \times d_{mid}}$ with columns sampled from the Gaussian $\mathcal{N}(0, K_g)$.

4. From $Z$, compute $K_g$ with given $\nu_h$

5. From $K_h$, sample the test dataset $Y \in \mathbb{R}^{D \times d_{out}}$ with columns sampled from the Gaussian $\mathcal{N}(0, K_h)$.

We utilized four AMD Opteron 6136 processors (2.4 GHz, 32 cores) and 128 GB of RAM to generate our synthetic dataset. The maximum possible dataset size for 128 GB of RAM is approximately 52,500; however, we opted for a synthetic dataset size of 22,000 due to the computational expense associated with sampling the Matérn kernel. This decision was made considering the time complexity of $\mathcal{O}(n^3)$ and the space complexity of $\mathcal{O}(n^2)$ involved. Out of the 22,000 dataset points, 20,000 were allocated for training data, and 2,000 were used for the test dataset

### A.1.2   Real-world dataset: WESAD

In our study, we utilized the Wearable Stress and Affect Detection (WESAD) dataset to train our AccNets for binary classification. The WESAD dataset, which is publicly accessible, provides multimodal physiological and motion data collected from 15 subjects using devices worn on the wrist and chest. For the purpose of our experiment, we specifically employed the Empatica E4 wrist device to distinguish between non-stress (baseline) and stress conditions, simplifying the classification task to these two categories.

After preprocessing, the dataset comprised a total of 136,482 instances. We implemented a train-test split ratio of approximately 75:25, resulting in 100,000 instances for the training set and 36,482 instances for the test set. The overall hyperparameters and architecture of the AccNets model applied to the WESAD dataset were largely consistent with those used for our synthetic data. The primary differences were the use of 100 epochs for each iteration of Ni from Ns, and a learning rate set to 1e-5.

---

[4]The code used for experiments are publicly available here

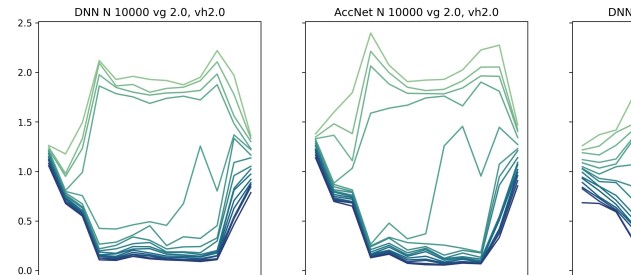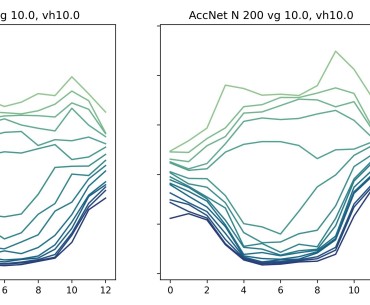

Figure 4: A comparison: singular values of the weight matrices for DNN and AccNets models. The first two plots represent cases where $N = 10000$ while the right two plots correspond to $N = 200$. The number of outliers at the top of each plot signifies the rank of each network. The plots with $N = 10000$ datasets demonstrate a clearer capture of the true rank compared to those with $N = 200$ indicating that a higher dataset count provides more accurate rank determination
.

## A.2 Model setups

To investigate the scaling law of test error for our synthetic data, we trained models using $N_i$ datapoints from our training data, where $N = [100, 200, 500, 1000, 2000, 5000, 10000, 20000]$. The models employed for this analysis included the kernel method, shallow networks, fully connected deep neural networks (FC DNN), and AccNets. For FC DNN and AccNets, we configured the network depth to 12 layers, with the layer widths set as $[d_{in}, 500, 500, ..., 500, d_{out}]$ for DNNs, and $[d_i n, 900, 100, 900, ..., 100, 900, d_{out}]$ for AccNets.

To ensure a comparable number of neurons, the width for the shallow networks was set to 50,000, resulting in dimensions of $[d_{in}, 50000, d_{out}]$.

We utilized ReLU as the activation function and $L^1$-norm as the cost function, with the Adam optimizer. The total number of batch was set to 5, and the training process was conducted over 3600 epochs, divided into three phases. The detailed optimizer parameters are as follows:

1. For the first 1200 epochs: learning rate $(lr) = 1.5 * 0.001$, weight decay $= 0$
2. For the second 1200 epochs: $lr = 0.4 * 0.001$, weight decay $= 0.002$
3. For the final 1200 epochs: $lr = 0.1 * 0.001$, weight decay $= 0.005$

We conducted experiments utilizing 12 NVIDIA V100 GPUs (each with 32GB of memory) over a period of 6.3 days to train the synthetic dataset. In contrast, training the WESAD dataset required only one hour on a single V100 GPU.

## A.3 Additional experiments

# B AccNet Generalization Bounds

The proof of generalization for shallow networks (Theorem 1) is the special case $L = 1$ of the proof of Theorem 2, so we only prove the second:

**Theorem 6.** *Consider an accordion net of depth $L$ and widths $d_L, \ldots, d_0$, with corresponding set of functions $\mathcal{F} = \{f_{L:1} : \|f_\ell\|_{F_1} \leq R_\ell, Lip(f_\ell) \leq \rho_\ell\}$ with input space $\Omega = B(0, r)$. For any $\rho$-Lipschitz loss function $\ell(x, f(x))$ with $|\ell(x, y)| \leq c_0$, we know that with probability $1 - \delta$ over the sampling of the training set $X$ from the distribution $\pi$, we have for all $f \in \mathcal{F}$*

$$\mathcal{L}(f) - \tilde{\mathcal{L}}_N(f) \leq C\rho_{L:1}r \sum_{\ell'=1}^{L} \frac{R_{\ell'}}{\rho_{\ell'}} \sqrt{d_{\ell'} + d_{\ell'-1}} \frac{\log N}{\sqrt{N}}(1 + o(1)) + c_0\sqrt{\frac{2\log 2/\delta}{N}}.$$

*Proof.* The strategy is: (1) prove a covering number bound on $\mathcal{F}$ (2) use it to obtain a Rademacher complexity bound, (3) use the Rademacher complexity to bound the generalization error.

(1) We define $f_\ell = V_\ell \circ \sigma \circ W_\ell$ so that $f_\theta = f_{L:1} = f_L \circ \cdots \circ f_1$. First notice that we can write each $f_\ell$ as convex combination of its neurons:

$$f_\ell(x) = \sum_{i=1}^{w_\ell} v_{\ell,i}\sigma(w_{\ell,i}^T x) = R_\ell \sum_{i=1}^{w_\ell} c_{\ell,i}\bar{v}_{\ell,i}\sigma(\bar{w}_{\ell,i}^T x)$$

for $\bar{w}_{\ell,i} = \frac{w_{\ell,i}}{\|w_{\ell,i}\|}$, $\bar{v}_{\ell,i} = \frac{v_{\ell,i}}{\|v_{\ell,i}\|}$, $R_\ell = \sum_{i=1}^{\ell} \|v_{\ell,i}\| \|w_{\ell,i}\|$ and $c_{\ell,i} = \frac{1}{R_\ell} \|v_{\ell,i}\| \|w_{\ell,i}\|$.

Let us now consider a sequence $\epsilon_k = 2^{-k}$ for $k = 0, \ldots, K$ and define $\tilde{v}_{\ell,i}^{(k)}, \tilde{w}_{\ell,i}^{(k)}$ to be the $\epsilon_k$-covers of $\bar{v}_{\ell,i}, \bar{w}_{\ell,i}$, furthermore we may choose $\tilde{v}_{\ell,i}^{(0)} = \tilde{w}_{\ell,i}^{(0)} = 0$ since every unit vector is within a $\epsilon_0 = 1$ distance of the origin. We will now show that on can approximate $f_\theta$ by approximating each of the $f_\ell$ by functions of the form

$$\tilde{f}_\ell(x) = R_\ell \sum_{k=1}^{K_\ell} \frac{1}{M_{k,\ell}} \sum_{m=1}^{M_{k,\ell}} \tilde{v}_{\ell,i_{\ell,m}^{(k)}}^{(k)} \sigma(\tilde{w}_{\ell,i_{\ell,m}^{(k)}}^{(k)T} x) - \tilde{v}_{\ell,i_{\ell,m}^{(k)}}^{(k-1)} \sigma(\tilde{w}_{\ell,i_{\ell,m}^{(k)}}^{(k-1)T} x)$$

for indices $i_{\ell,m}^{(k)} = 1, \ldots, w_\ell$ choosen adequately. Notice that the number of functions of this type equals the number of $M_{k,\ell}$ quadruples $(\tilde{v}_{\ell,i_{\ell,m}^{(k)}}^{(k)}, \tilde{w}_{\ell,i_{\ell,m}^{(k)}}^{(k)T}, \tilde{v}_{\ell,i_{\ell,m}^{(k)}}^{(k-1)}, \tilde{w}_{\ell,i_{\ell,m}^{(k)}}^{(k-1)T})$ where these vectors belong to the $\epsilon_k$- resp. $\epsilon_{k-1}$-coverings of the $d_{in}$- resp. $d_{out}$-dimensional unit sphere. Thus the number of such functions is bounded by

$$\prod_{k=1}^{K_\ell} \left( \mathcal{N}_2(\mathbb{S}^{d_{in}-1}, \epsilon_k)\mathcal{N}_2(\mathbb{S}^{d_{out}-1}, \epsilon_k)\mathcal{N}_2(\mathbb{S}^{d_{in}-1}, \epsilon_{k-1})\mathcal{N}_2(\mathbb{S}^{d_{out}-1}, \epsilon_{k-1}) \right)^{M_{k,\ell}},$$

and we have this choice for all $\ell = 1, \ldots, L$. We will show that with sufficiently large $M_{k,\ell}$ this set of functions $\epsilon$-covers $\mathcal{F}$ which then implies that

$$\log \mathcal{N}_2(\mathcal{F}, \epsilon) \leq 2 \sum_{\ell=1}^{L} \sum_{k=1}^{K_\ell} M_{k,\ell} \left( \log \mathcal{N}_2(\mathbb{S}^{d_{in}-1}, \epsilon_{k-1}) + \log \mathcal{N}_2(\mathbb{S}^{d_{in}-1}, \epsilon_{k-1}) \right).$$

We will use the probabilistic method to find the right indices $i_{\ell,m}^{(k)}$ to approximate a function $f_\ell = R_\ell \sum_{i=1}^{w_\ell} c_{\ell,i}\bar{v}_{\ell,i}\sigma(\bar{w}_{\ell,i}^T x)$ with a function $\tilde{f}_\ell$. We take all $i_{\ell,m}^{(k)}$ to be i.i.d. equal to the index $i = 1, \cdots, w_\ell$ with probability $c_{\ell,i}$, so that in expectation

$$\mathbb{E}\tilde{f}_\ell(x) = R_\ell \sum_{k=1}^{K_\ell} \sum_{i=1}^{w_\ell} c_{\ell,i} \left( \tilde{v}_{\ell,i}^{(k)}\sigma(\tilde{w}_{\ell,i}^{(k)T} x) - \tilde{v}_{\ell,i}^{(k-1)}\sigma(\tilde{w}_{\ell,i}^{(k-1)T} x) \right)$$

$$= R_\ell \sum_{i=1}^{w_\ell} c_{\ell,i}\tilde{v}_{\ell,i}^{(K)}\sigma(\tilde{w}_{\ell,i}^{(K)T} x).$$

We will show that this expectation is $O(\epsilon_{K_\ell})$-close to $f_\ell$ and that the variance of $\tilde{f}_\ell$ goes to zero as the $M_{\ell,k}$s grow, allowing us to bound the expected error $\mathbb{E} \left\| f_{L:1} - \tilde{f}_{L:1} \right\|_\pi^2 \leq \epsilon^2$ which then implies that there must be at least one choice of indices $i_{\ell,m}^{(k)}$ such that $\left\| f_{L:1} - \tilde{f}_{L:1} \right\|_\pi \leq \epsilon$.

547 Let us first bound the distance

$$
\begin{aligned}
\left\| f_\ell(x) - \mathbb{E}\tilde{f}_\ell(x) \right\| &= R_\ell \left\| \sum_{i=1}^{w_\ell} c_{\ell,i} \left( \bar{v}_{\ell,i} \sigma(\bar{w}_{\ell,i}^T x) - \tilde{v}_{\ell,i}^{(K)} \sigma(\tilde{w}_{\ell,i}^{(K)T} x) \right) \right\| \\
&\leq R_\ell \sum_{i=1}^{w_\ell} c_{\ell,i} \left( \left\| \left( \bar{v}_{\ell,i} - \tilde{v}_{\ell,i}^{(K)} \right) \sigma(\bar{w}_{\ell,i}^T x) \right\| + \left\| \tilde{v}_{\ell,i}^{(K)} \left( \sigma(\bar{w}_{\ell,i}^T x) - \sigma(\tilde{w}_{\ell,i}^{(K)T} x) \right) \right\| \right) \\
&\leq R_\ell \sum_{i=1}^{w_\ell} c_{\ell,i} \left( \left\| \bar{v}_{\ell,i} - \tilde{v}_{\ell,i}^{(K)} \right\| \left\| \bar{w}_{\ell,i}^T x \right\| + \left\| \tilde{v}_{\ell,i}^{(K)} \right\| \left\| \bar{w}_{\ell,i}^T x - \tilde{w}_{\ell,i}^{(K)T} x \right\| \right) \\
&\leq 2R_\ell \sum_{i=1}^{w_\ell} c_{\ell,i} \epsilon_{K_\ell} \left\| x \right\| \\
&= 2R_\ell \epsilon_{K_\ell} \left\| x \right\|.
\end{aligned}
$$

548 Then we bound the trace of the covariance of $\tilde{f}_\ell$ which equals the expected square distance between
549 $\tilde{f}_\ell$ and its expectation:

$$
\begin{aligned}
&\mathbb{E} \left\| \tilde{f}_\ell(x) - \mathbb{E}\tilde{f}_\ell(x) \right\|^2 \\
&= \sum_{k=1}^{K_\ell} \frac{R_\ell^2}{M_{k,\ell}^2} \sum_{m=1}^{M_{k,\ell}} \mathbb{E} \left\| \tilde{v}_{\ell,i_{\ell,m}^{(k)}}^{(k)} \sigma(\tilde{w}_{\ell,i_{\ell,m}^{(k)}}^{(k)T} x) - \tilde{v}_{\ell,i_{\ell,m}^{(k)}}^{(k-1)} \sigma(\tilde{w}_{\ell,i_{\ell,m}^{(k)}}^{(k-1)T} x) - \mathbb{E} \left[ \tilde{v}_{\ell,i_{\ell,m}^{(k)}}^{(k)} \sigma(\tilde{w}_{\ell,i_{\ell,m}^{(k)}}^{(k)T} x) - \tilde{v}_{\ell,i_{\ell,m}^{(k)}}^{(k-1)} \sigma(\tilde{w}_{\ell,i_{\ell,m}^{(k)}}^{(k-1)T} x) \right] \right\|^2 \\
&\leq \sum_{k=1}^{K_\ell} \frac{R_\ell^2}{M_{k,\ell}^2} \sum_{m=1}^{M_{k,\ell}} \mathbb{E} \left\| \tilde{v}_{\ell,m}^{(k)} \sigma(\tilde{w}_{\ell,m}^{(k)T} x) - \tilde{v}_{\ell,m}^{(k-1)} \sigma(\tilde{w}_{\ell,m}^{(k-1)T} x) \right\|^2 \\
&= \sum_{k=1}^{K_\ell} \frac{R_\ell^2}{M_{k,\ell}} \sum_{i=1}^{w_\ell} c_i \left\| \tilde{v}_{\ell,i}^{(k)} \sigma\left( \tilde{w}_{\ell,i}^{(k)T} x \right) - \tilde{v}_{\ell,i}^{(k-1)} \sigma\left( \tilde{w}_{\ell,i}^{(k-1)T} x \right) \right\|^2 \\
&\leq \sum_{k=1}^{K_\ell} \frac{2R_\ell^2 \|x\|^2}{M_{k,\ell}} \sum_{i=1}^{w_\ell} c_i \left\| \tilde{v}_{\ell,i}^{(k)} \right\|^2 \left\| \tilde{w}_{\ell,i}^{(k)} - \tilde{w}_{\ell,i}^{(k-1)} \right\|^2 + c_i \left\| \tilde{v}_{\ell,i}^{(k)} - \tilde{v}_{\ell,i}^{(k-1)} \right\|^2 \left\| \tilde{w}_{\ell,i}^{(k-1)} \right\|^2 \\
&\leq \sum_{k=1}^{K_\ell} \frac{4R_\ell^2 \|x\|^2}{M_{k,\ell}} (\epsilon_k + \epsilon_{k-1})^2 \\
&\leq \sum_{k=1}^{K_\ell} \frac{36 R_\ell^2 \|x\|^2}{M_{k,\ell}} \epsilon_k^2.
\end{aligned}
$$

550 Putting both together, we obtain

$$
\begin{aligned}
\mathbb{E} \left\| f_\ell(x) - \tilde{f}_\ell(x) \right\|^2 &\leq 4R_\ell^2 \epsilon_{K_\ell}^2 \|x\|^2 + \sum_{k=1}^{K_\ell} \frac{36 R_\ell^2 \|x\|^2}{M_{k,\ell}} \epsilon_k^2 \\
&= 4R_\ell^2 \|x\|^2 \left( \epsilon_{K_\ell}^2 + 9 \sum_{k=1}^{K_\ell} \frac{\epsilon_k^2}{M_{k,\ell}} \right).
\end{aligned}
$$

551 We will now use this bound, together with the Lipschitzness of $f_\ell$ to bound the error
552 $\mathbb{E} \left\| f_{L:1}(x) - \tilde{f}_{L:1}(x) \right\|^2$. We will do this by induction on the distances $\mathbb{E} \left\| f_{\ell:1}(x) - \tilde{f}_{\ell:1}(x) \right\|^2$.
553 We start by

$$
\mathbb{E} \left\| f_1(x) - \tilde{f}_1(x) \right\|^2 \leq 4R_1^2 \|x\|^2 \left( \epsilon_{K_\ell}^2 + 9 \sum_{k=1}^{K_\ell} \frac{\epsilon_k^2}{M_{k,1}} \right).
$$

554 And for the induction step, we condition on the layers $f_{\ell-1:1}$

$$\mathbb{E}\left\|f_{\ell:1}(x) - \tilde{f}_{\ell:1}(x)\right\|^2 = \mathbb{E}\left[\mathbb{E}\left[\left\|f_{\ell:1}(x) - \tilde{f}_{\ell:1}(x)\right\|^2 | \tilde{f}_{\ell-1:1}\right]\right]$$

$$= \mathbb{E}\left\|f_{\ell:1}(x) - \mathbb{E}\left[\tilde{f}_{\ell:1}(x) | \tilde{f}_{\ell-1:1}\right]\right\|^2 + \mathbb{E}\mathbb{E}\left[\left\|\tilde{f}_{\ell:1}(x) - \mathbb{E}\left[\tilde{f}_{\ell:1}(x) | \tilde{f}_{\ell-1:1}\right]\right\|^2 | \tilde{f}_{\ell-1:1}\right]$$

$$= \mathbb{E}\left\|f_{\ell:1}(x) - f_\ell(\tilde{f}_{\ell-1:1}(x))\right\|^2 + \mathbb{E}\mathbb{E}\left[\left\|\tilde{f}_{\ell:1}(x) - f_\ell(\tilde{f}_{\ell-1:1}(x))\right\|^2 | \tilde{f}_{\ell-1:1}\right]$$

$$\leq \rho_\ell^2 \mathbb{E}\left\|f_{\ell-1:1}(x) - \tilde{f}_{\ell-1:1}(x)\right\|^2 + 4R_\ell^2 \mathbb{E}\left\|\tilde{f}_{\ell-1:1}(x)\right\|^2 \left(\epsilon_{K_\ell}^2 + 9\sum_{k=1}^{K_\ell}\frac{\epsilon_k^2}{M_{k,\ell}}\right).$$

555 Now since

$$\mathbb{E}\left\|\tilde{f}_{\ell-1:1}(x)\right\|^2 \leq \|f_{\ell-1:1}(x)\|^2 + \mathbb{E}\left\|f_{\ell-1:1}(x) - \tilde{f}_{\ell-1:1}(x)\right\|^2$$

$$\leq \rho_{\ell-1}^2\cdots\rho_1^2 \|x\|^2 + \mathbb{E}\left\|f_{\ell-1:1}(x) - \tilde{f}_{\ell-1:1}(x)\right\|^2$$

556 we obtain that

$$\mathbb{E}\left\|f_{\ell:1}(x) - \tilde{f}_{\ell:1}(x)\right\|^2 \leq \left(\rho_\ell^2 + 4R_\ell^2\left(\epsilon_{K_\ell}^2 + 9\sum_{k=1}^{K_\ell}\frac{\epsilon_k^2}{M_{k,\ell}}\right)\right)\mathbb{E}\left\|f_{\ell-1:1}(x) - \tilde{f}_{\ell-1:1}(x)\right\|^2$$

$$+ 4R_\ell^2\rho_{\ell-1}^2\cdots\rho_1^2 \|x\|^2 \left(\epsilon_{K_\ell}^2 + 9\sum_{k=1}^{K_\ell}\frac{\epsilon_k^2}{M_{k,\ell}}\right).$$

557 We define $\tilde{\rho}_\ell^2 = \rho_\ell^2\left[1 + 4\frac{R_\ell^2}{\rho_\ell^2}\left(\epsilon_{K_\ell}^2 + 9\sum_{k=1}^{K_\ell}\frac{\epsilon_k^2}{M_{k,\ell}}\right)\right]$ and obtain

$$\mathbb{E}\left\|f_{L:1}(x) - \tilde{f}_{L:1}(x)\right\|^2 \leq 4\sum_{\ell=1}^{L}\tilde{\rho}_{L:\ell+1}^2 R_\ell^2\rho_{\ell-1:1}^2 \|x\|^2 \left(\epsilon_{K_\ell}^2 + 9\sum_{k=1}^{K_\ell}\frac{\epsilon_k^2}{M_{k,\ell}}\right).$$

558 Thus for any distribution $\pi$ over the ball $B(0,r)$, there is a choice of indices $i_{\ell,m}^{(k)}$ such that

$$\left\|f_{L:1} - \tilde{f}_{L:1}\right\|_\pi^2 \leq 4\sum_{\ell=1}^{L}\tilde{\rho}_{L:\ell+1}^2 R_\ell^2\rho_{\ell-1:1}^2 r^2 \left(\epsilon_{K_\ell}^2 + 9\sum_{k=1}^{K_\ell}\frac{\epsilon_k^2}{M_{k,\ell}}\right).$$

559 We now simply need to choose $K_\ell$ and $M_{k,\ell}$ adequately. To reach an error of $2\epsilon$, we choose

$$K_\ell = \left\lceil -\log\epsilon + \frac{1}{2}\log\left[4\rho_{L:1}^2 r^2\left(\sum_{\ell'=1}^{L}\frac{R_{\ell'}}{\rho_{\ell'}}\sqrt{d_{\ell'}+d_{\ell'-1}}\right)\frac{R_\ell}{\rho_\ell\sqrt{d_\ell+d_{\ell-1}}}\right]\right\rceil$$

560 where $\rho_{L:1} = \prod_{\ell=1}^{L}\rho_\ell$. Notice that that $\epsilon_{K_\ell}^2 \leq \frac{1}{4\rho_{L:1}^2 r^2\left(\sum_{\ell'=1}^{L}\frac{R_{\ell'}}{\rho_{\ell'}}\sqrt{d_{\ell'}+d_{\ell'-1}}\right)}\frac{\rho_\ell\sqrt{d_\ell+d_{\ell-1}}}{R_\ell}\epsilon^2$.

561 Given $s_0 = \sum_{k=1}^{\infty}\sqrt{k}2^{-k} \approx 1.3473 < \infty$, we define

$$M_{k,\ell} = \left\lceil 36\rho_{L:1}^2 r^2 s_0\left(\sum_{\ell'=1}^{L}\frac{R_{\ell'}}{\rho_{\ell'}}\sqrt{d_{\ell'}+d_{\ell'-1}}\right)\frac{R_\ell}{\rho_\ell\sqrt{d_\ell+d_{\ell-1}}}\frac{2^{-k}}{\sqrt{k}}\frac{1}{\epsilon^2}\right\rceil.$$

562  So that for all $\ell$

$$4\frac{R_\ell^2}{\rho_\ell^2}\left(\epsilon_{K_\ell}^2 + 9\sum_{k=1}^{K_\ell}\frac{\epsilon_k^2}{M_{k,\ell}}\right) \leq \frac{\frac{R_\ell}{\rho_\ell}\sqrt{d_\ell + d_{\ell-1}}}{\rho_{L:1}^2 r^2\left(\sum_{\ell'=1}^{L}\frac{R_\ell}{\rho_\ell}\sqrt{d_\ell + d_{\ell-1}}\right)}\epsilon^2$$

$$+ \frac{\frac{R_\ell}{\rho_\ell}\sqrt{d_\ell + d_{\ell-1}}}{\rho_{L:1}^2 r^2\left(\sum_{\ell'=1}^{L}\frac{R_\ell}{\rho_\ell}\sqrt{d_\ell + d_{\ell-1}}\right)}\epsilon^2\frac{\sum_{k'=1}^{K_\ell}\sqrt{k'}2^{-k'}}{s_0}$$

$$\leq 2\frac{\frac{R_\ell}{\rho_\ell}\sqrt{d_\ell + d_{\ell-1}}}{\rho_{L:1}^2 r^2\left(\sum_{\ell'=1}^{L}\frac{R_\ell}{\rho_\ell}\sqrt{d_\ell + d_{\ell-1}}\right)}\epsilon^2.$$

563  Now this also implies that

$$\tilde{\rho}_\ell \leq \rho_\ell \exp\left(2\frac{\frac{R_\ell}{\rho_\ell}\sqrt{d_\ell + d_{\ell-1}}}{\rho_{L:1}^2 r^2\left(\sum_{\ell'=1}^{L}\frac{R_\ell}{\rho_\ell}\sqrt{d_\ell + d_{\ell-1}}\right)}\epsilon^2\right)$$

564  and thus

$$\tilde{\rho}_{L:\ell+1} \leq \rho_{L:\ell+1}\exp\left(2\frac{\sum_{\ell'=\ell+1}^{L}\frac{R_\ell}{\rho_\ell}\sqrt{d_\ell + d_{\ell-1}}}{\rho_{L:1}^2 r^2\left(\sum_{\ell'=1}^{L}\frac{R_\ell}{\rho_\ell}\sqrt{d_\ell + d_{\ell-1}}\right)}\epsilon^2\right) \leq \rho_{L:\ell+1}\exp\left(\frac{2}{\rho_{L:1}^2 r^2}\epsilon^2\right).$$

565  Putting it all together, we obtain that

$$\left\|f_{L:1} - \tilde{f}_{L:1}\right\|_\pi^2 \leq 4\sum_{\ell=1}^{L}\tilde{\rho}_{L:\ell+1}^2 R_\ell^2 \rho_{\ell-1:1}^2 r^2\left(\epsilon_{K_\ell}^2 + 9\sum_{k=1}^{K_\ell}\frac{\epsilon_k^2}{M_{k,\ell}}\right)$$

$$\leq \exp\left(\frac{2}{\rho_{L:1}^2 r^2}\epsilon^2\right)\rho_{L:1}^2 r^2\sum_{\ell=1}^{L}4\frac{R_\ell^2}{\rho_\ell^2}\left(\epsilon_{K_\ell}^2 + 9\sum_{k=1}^{K_\ell}\frac{\epsilon_k^2}{M_{k,\ell}}\right)$$

$$\leq 2\exp\left(\frac{2}{\rho_{L:1}^2 r^2}\epsilon^2\right)\epsilon^2$$

$$= 2\epsilon^2 + O(\epsilon^4).$$

566  Now since $\log\mathcal{N}_2(\mathbb{S}^{d_\ell-1}, \epsilon) = d_\ell \log\left(\frac{1}{\epsilon} + 1\right)$ and

$$M_{k,\ell} \leq 36\rho_{L:1}^2 r^2 s_0\left(\sum_{\ell'=1}^{L}\frac{R_{\ell'}}{\rho_{\ell'}}\sqrt{d_{\ell'} + d_{\ell'-1}}\right)\frac{R_\ell}{\rho_\ell\sqrt{d_\ell + d_{\ell-1}}}\frac{2^{-k}}{\sqrt{k}}\frac{1}{\epsilon^2} + 1,$$

567  we have

$$\log\mathcal{N}_2\left(\mathcal{F}, \sqrt{2}\exp\left(\frac{\epsilon^2}{\rho_{L:1}^2 r^2}\right)\epsilon\right) \leq 2\sum_{\ell=1}^{L}\sum_{k=1}^{K_\ell}M_{k,\ell}\left(\log\mathcal{N}_2(\mathbb{S}^{d_\ell-1}, \epsilon_{k-1}) + \log\mathcal{N}_2(\mathbb{S}^{d_{\ell-1}-1}, \epsilon_{k-1})\right)$$

$$\leq 2\sum_{\ell=1}^{L}\sum_{k=1}^{K_\ell}M_{k,\ell}\left(d_\ell + d_{\ell-1}\right)\log(\frac{1}{\epsilon_{k-1}} + 1)$$

$$\leq 72s_0\rho_{L:1}^2 r^2\left(\sum_{\ell'=1}^{L}\frac{R_{\ell'}}{\rho_{\ell'}}\sqrt{d_{\ell'} + d_{\ell'-1}}\right)\sum_{\ell=1}^{L}\frac{R_\ell}{\rho_\ell}\sqrt{d_\ell + d_{\ell-1}}\sum_{k=1}^{K_\ell}\frac{2^{-k}\log(\frac{1}{\epsilon_{k-1}} + 1)}{\sqrt{k}}\frac{1}{\epsilon^2}$$

$$+ 2\sum_{\ell=1}^{L}(d_\ell + d_{\ell-1})\sum_{k=1}^{K_\ell}\log(\frac{1}{\epsilon_{k-1}} + 1)$$

$$\leq 72s_0^2\rho_{L:1}^2 r^2\left(\sum_{\ell'=1}^{L}\frac{R_{\ell'}}{\rho_{\ell'}}\sqrt{d_{\ell'} + d_{\ell'-1}}\right)^2\frac{1}{\epsilon^2} + o(\epsilon^{-2}).$$

568 The diameter of $\mathcal{F}$ is smaller than $\rho_{L:1}r$, so for all $\delta \geq \rho_{L:1}r$, $\log \mathcal{N}_2(\mathcal{F}, \delta) = 0$. For all $\delta \leq \rho_{L:1}r$
569 we choose $\epsilon = \frac{\delta}{\sqrt{2e}}$ so that $\sqrt{2}\exp\left(\frac{\epsilon^2}{\rho_{L:1}^2 r^2}\right)\epsilon \leq \delta$ and therefore

$$\log \mathcal{N}_2(\mathcal{F}, \delta) \leq 144 s_0^2 e \rho_{L:1}^2 r^2 \left(\sum_{\ell'=1}^{L} \frac{R_{\ell'}}{\rho_{\ell'}}\sqrt{d_{\ell'}+d_{\ell'-1}}\right)^2 \frac{1}{\delta^2} + o(\delta^{-2}).$$

570 (2) Our goal now is to use chaining / Dudley's theorem to bound the Rademacher complexity
571 $R(\mathcal{F}(X))$ evaluated on a set $X$ of size $N$ (e.g. Lemma 27.4 in [Understanding Machine Learning])
572 of our set:

573 **Lemma 7.** *Let* $c = \max_{f\in\mathcal{F}} \frac{1}{\sqrt{N}}\|f(X)\|$, *then for any integer* $M > 0$,

$$R(\mathcal{F}(X)) \leq c2^{-M} + \frac{6c}{\sqrt{N}}\sum_{k=1}^{M} 2^{-k}\sqrt{\log \mathcal{N}(\mathcal{F}, c2^{-k})}.$$

574 To apply it to our setting, first note that for all $x \in B(0, r)$, $\|f_{L:1}(x)\| \leq \rho_{L:1}r$ so that $c =$
575 $\max_{f\in\mathcal{F}}\frac{1}{\sqrt{N}}\|f(X)\| \leq \rho_{L:1}r$, we then have

$$R(\mathcal{F}(X)) \leq c2^{-M} + \frac{6c}{\sqrt{N}}\sum_{k=1}^{M} 2^{-k}12s_0\sqrt{e}\rho_{L:1}r\sum_{\ell'=1}^{L}\frac{R_{\ell'}}{\rho_{\ell'}}\sqrt{d_{\ell'}+d_{\ell'-1}}c^{-1}2^k(1+o(1))$$

$$= c2^{-M} + \frac{72}{\sqrt{N}}Ms_0\sqrt{e}\rho_{L:1}r\sum_{\ell'=1}^{L}\frac{R_{\ell'}}{\rho_{\ell'}}\sqrt{d_{\ell'}+d_{\ell'-1}}(1+o(1)).$$

576 Taking $M = \left\lceil -\log_2\left(\frac{72}{\sqrt{N}}s_0\sqrt{e}\sum_{\ell'=1}^{L}\frac{R_{\ell'}}{\rho_{\ell'}}\sqrt{d_{\ell'}+d_{\ell'-1}}\right)\right\rceil$, we obtain

$$R(\mathcal{F}(X)) \leq \frac{72}{\sqrt{N}}Ms_0\sqrt{e}\rho_{L:1}r\sum_{\ell'=1}^{L}\frac{R_{\ell'}}{\rho_{\ell'}}\sqrt{d_{\ell'}+d_{\ell'-1}}(1+M(1+o(1)))$$

$$\leq \frac{144}{\sqrt{N}}Ms_0\sqrt{e}\rho_{L:1}r\sum_{\ell'=1}^{L}\frac{R_{\ell'}}{\rho_{\ell'}}\sqrt{d_{\ell'}+d_{\ell'-1}}\left\lceil -\log_2\left(\frac{72}{\sqrt{N}}s_0\sqrt{e}\sum_{\ell'=1}^{L}\frac{R_{\ell'}}{\rho_{\ell'}}\sqrt{d_{\ell'}+d_{\ell'-1}}\right)\right\rceil(1+o(1))$$

$$\leq C\rho_{L:1}r\sum_{\ell'=1}^{L}\frac{R_{\ell'}}{\rho_{\ell'}}\sqrt{d_{\ell'}+d_{\ell'-1}}\frac{\log N}{\sqrt{N}}(1+o(1)).$$

577 (3) For any $\rho$-Lipschitz loss function $\ell(x, f(x))$ with $|\ell(x, y)| \leq c_0$, we know that with probability
578 $1 - \delta$ over the sampling of the training set $X$ from the distribution $\pi$, we have for all $f \in \mathcal{F}$

$$\mathbb{E}_{x\sim\pi}\left[\ell(x, f(x))\right] - \frac{1}{N}\sum_{i=1}^{N}\ell(x_i, f(x_i)) \leq 2\mathbb{E}_{X'}\left[R(\ell\circ\mathcal{F}(X'))\right] + c_0\sqrt{\frac{2\log^2/\delta}{N}}$$

$$\leq 2C\rho_{L:1}r\sum_{\ell'=1}^{L}\frac{R_{\ell'}}{\rho_{\ell'}}\sqrt{d_{\ell'}+d_{\ell'-1}}\frac{\log N}{\sqrt{N}}(1+o(1)) + c_0\sqrt{\frac{2\log^2/\delta}{N}}.$$

579 $\qquad\qquad\qquad\qquad\qquad\qquad\qquad\qquad\qquad\qquad\qquad\qquad\qquad\qquad\qquad\qquad\qquad\qquad\qquad\square$

# C Composition of Sobolev Balls

581 **Proposition 8** (Proposition 3 from the main.)**.** *Given a distribution* $\pi$ *with support in* $B(0, r)$, *we*
582 *have that with probability* $1 - \delta$ *for all functions* $f \in \mathcal{F} = \{f : \|f\|_{W^{\nu,2}} \leq R, \|f\|_\infty \leq R\}$

$$\mathcal{L}(f) - \tilde{\mathcal{L}}_N(f) \leq 2C_1 RE_{\nu/d}(N) + c_0\sqrt{\frac{2\log^2/\delta}{N}}.$$

583 *where* $E_r(N) = N^{-\frac{1}{2}}$ *if* $r > \frac{1}{2}$, $E_r(N) = N^{-\frac{1}{2}}\log N$ *if* $r = \frac{1}{2}$, *and* $E_r(N) = N^{-r}$ *if* $r < \frac{1}{2}$.

*Proof.* (1) We know from Theorem 5.2 of [9] that the Sobolev ball $B_{W^{\nu,2}}(0, R)$ over any $d$-dimensional hypercube $\Omega$ satisfies

$$\log \mathcal{N}_2(B_{W^{\nu,2}}(0, R), \pi, \epsilon) \leq C_0 \left( \frac{R}{\epsilon} \right)^{\frac{d}{\nu}}$$

for a constant $c$ and any measure $\pi$ supported in the hypercube.

(2) By Dudley's theorem we can bound the Rademacher complexity of our function class $\mathcal{B}(X)$ evaluated on any training set $X$:

$$R(\mathcal{B}(X)) \leq R2^{-M} + \frac{6R}{\sqrt{N}} \sum_{k=1}^{M} 2^{-k} \sqrt{C_0 \left( \frac{R}{R2^{-k}} \right)^{\frac{d}{\nu}}}$$

$$= R2^{-M} + \frac{6R}{\sqrt{N}} \sqrt{C_0} \sum_{k=1}^{M} 2^{k(\frac{d}{2\nu}-1)}.$$

If $2\nu = d$, we take $M = \frac{1}{2} \log N$ and obtain the bound

$$\frac{R}{\sqrt{N}} + \frac{6R}{\sqrt{N}} \sqrt{C_0} \frac{1}{2} \log N \leq C_1 R \frac{\log N}{\sqrt{N}}.$$

If $2\nu > d$, we take $M = \infty$ and obtain the bound

$$\frac{6R}{\sqrt{N}} \sqrt{C_0} \left( \frac{2^{\frac{d}{2\nu}-1}}{1 - 2^{\frac{d}{2\nu}-1}} \right) \leq C_1 R \frac{1}{\sqrt{N}}.$$

If $2\nu < d$, we take $M = \frac{\nu}{d} \log N$ and obtain the bound

$$R2^{-M} + \frac{6R}{\sqrt{N}} \sqrt{C_0} 2^{\frac{d}{2\nu}-1} \left( \frac{2^{M(\frac{d}{2\nu}-1)} - 1}{2^{\frac{d}{2\nu}-1} - 1} \right) \leq C_1 R N^{-\frac{\nu}{d}}.$$

Putting it all together, we obtain that $R(\mathcal{B}(X)) \leq C_1 E_{\nu/d}(N)$.

(3) For any $\rho$-Lipschitz loss function $\ell(x, f(x))$ with $|\ell(x, y)| \leq c_0$, we know that with probability $1 - \delta$ over the sampling of the training set $X$ from the distribution $\pi$, we have for all $f \in \mathcal{F}$

$$\mathbb{E}_{x \sim \pi} \left[ \ell(x, f(x)) \right] - \frac{1}{N} \sum_{i=1}^{N} \ell(x_i, f(x_i)) \leq 2 \mathbb{E}_{X'} \left[ R(\ell \circ \mathcal{F}(X')) \right] + c_0 \sqrt{\frac{2 \log^{2}/\delta}{N}}$$

$$\leq 2 C_1 E_{\nu/d}(N) + c_0 \sqrt{\frac{2 \log^{2}/\delta}{N}}.$$

$\square$

**Proposition 9.** *Let $\mathcal{F}_1, \ldots, \mathcal{F}_L$ be set of functions mapping through the sets $\Omega_0, \ldots, \Omega_L$, then if all functions in $\mathcal{F}_\ell$ are $\rho_\ell$-Lipschitz, we have*

$$\log \mathcal{N}_2(\mathcal{F}_L \circ \cdots \circ \mathcal{F}_1, \sum_{\ell=1}^{L} \rho_{L:\ell+1} \epsilon_\ell) \leq \sum_{\ell=1}^{L} \log \mathcal{N}_2(\mathcal{F}_\ell, \epsilon_\ell).$$

*Proof.* For any distribution $\pi_0$ on $\Omega$ there is a $\epsilon_1$-covering $\tilde{\mathcal{F}}_1$ of $\mathcal{F}_1$ with $\left| \tilde{\mathcal{F}}_1 \right| \leq \mathcal{N}_2(\mathcal{F}_1, \epsilon_1)$ then for any $\tilde{f}_1 \in \tilde{\mathcal{F}}_1$ we choose a $\epsilon_2$-covering $\tilde{\mathcal{F}}_2$ w.r.t. the measure $\pi_1$ which is the measure of $f_1(x)$ if $x \sim \pi_0$ of $\mathcal{F}_2$ with $\left| \tilde{\mathcal{F}}_2 \right| \leq \mathcal{N}_2(\mathcal{F}_2, \epsilon)$, and so on until we obtain coverings for all $\ell$. Then the set $\tilde{\mathcal{F}} = \left\{ \tilde{f}_L \circ \cdots \circ \tilde{f}_1 : \tilde{f}_1 \in \tilde{\mathcal{F}}_1, \ldots, \tilde{f}_L \in \tilde{\mathcal{F}}_L \right\}$ is a $\sum_{\ell=1}^{L} \rho_{L:\ell+1} \epsilon_\ell$-covering of $\mathcal{F} = \mathcal{F}_L \circ \cdots \circ \mathcal{F}_1$,

indeed for any $f = f_{L:1}$ we choose $\tilde{f}_1 \in \tilde{\mathcal{F}}_1, \ldots, \tilde{f}_L \in \tilde{\mathcal{F}}_L$ that cover $f_1, \ldots, f_L$, then $\tilde{f}_{L:1}$ covers $f_{L:1}$:

$$\left\| f_{L:1} - \tilde{f}_{L:1} \right\|_\pi \le \sum_{\ell=1}^{L} \left\| f_{L:\ell} \circ \tilde{f}_{\ell-1:1} - f_{L:\ell+1} \circ \tilde{f}_{\ell:1} \right\|_\pi$$

$$\le \sum_{\ell=1}^{L} \left\| f_{L:\ell} - f_{L:\ell+1} \circ \tilde{f}_\ell \right\|_{\pi_{\ell-1}}$$

$$\le \sum_{\ell=1}^{L} \rho_{L:\ell+1} \epsilon_\ell,$$

and log cardinality of the set $\tilde{\mathcal{F}}$ is bounded $\sum_{\ell=1}^{L} \log \mathcal{N}_2(\mathcal{F}_\ell, \epsilon_\ell)$. $\qquad\square$

**Theorem 10.** *Let* $\mathcal{F} = \mathcal{F}_L \circ \cdots \circ \mathcal{F}_1$ *where* $\mathcal{F}_\ell = \left\{ f_\ell : \mathbb{R}^{d_{\ell-1}} \to \mathbb{R}^{d_\ell} \text{ s.t. } \|f_\ell\|_{W^{\nu_\ell,2}} \le R_\ell, \|f_\ell\|_\infty \le b_\ell, Lip(f_\ell) \le \rho_\ell \right\}$, *and let* $r^* = \min_\ell r_\ell$ *for* $r_\ell = \frac{\nu_\ell}{d_{\ell-1}}$, *then with probability* $1 - \delta$ *we have for all* $f \in \mathcal{F}$

$$\mathcal{L}(f) - \tilde{\mathcal{L}}_N(f) \le \rho C_0 \left( \sum_{\ell=1}^{L} (C_\ell \rho_{L:\ell+1} R_\ell)^{\frac{1}{r^*+1}} \right)^{r^*+1} E_{r^*}(N) + c_0 \sqrt{\frac{2 \log 2/\delta}{N}},$$

*where* $C_\ell$ *depends only on* $d_{\ell-1}, d_\ell, \nu_\ell, b_{\ell-1}$.

*Proof.* (1) We know from Theorem 5.2 of [9] that the Sobolev ball $B_{W^{\nu_\ell,2}}(0, R_\ell)$ over any $d_\ell$-dimensional hypercube $\Omega$ satisfies

$$\log \mathcal{N}_2(B_{W^{\nu,2}}(0, R_\ell), \pi_{\ell-1}, \epsilon_\ell) \le \left( C_\ell \frac{R_\ell}{\epsilon_\ell} \right)^{\frac{1}{r_\ell}}$$

for a constant $C_\ell$ that depends on the size of hypercube and the dimension $d_\ell$ and the regularity $\nu_\ell$ and any measure $\pi_{\ell-1}$ supported in the hypercube.

Thus Proposition 9 tells us that the composition of the Sobolev balls satisfies

$$\log \mathcal{N}_2(\mathcal{F}_L \circ \cdots \circ \mathcal{F}_1, \sum_{\ell=1}^{L} \rho_{L:\ell+1} \epsilon_\ell) \le \sum_{\ell=1}^{L} \left( C_\ell \frac{R_\ell}{\epsilon_\ell} \right)^{\frac{1}{r_\ell}}.$$

Given $r^* = \min_\ell r_\ell$, we can bound it by $\sum_{\ell=1}^{L} \left( C_\ell \frac{R_\ell}{\epsilon_\ell} \right)^{\frac{1}{r^*}}$ and by then choosing $\epsilon_\ell = \frac{\rho_{L:\ell+1}^{-1}(\rho_{L:\ell+1} C_\ell R_\ell)^{\frac{1}{r^*+1}}}{\sum_\ell (\rho_{L:\ell+1} C_\ell R_\ell)^{\frac{1}{r^*+1}}} \epsilon$, we obtain that

$$\log \mathcal{N}_2(\mathcal{F}_L \circ \cdots \circ \mathcal{F}_1, \epsilon) \le \left( \sum_{\ell=1}^{L} (\rho_{L:\ell+1} C_\ell R_\ell)^{\frac{1}{r^*+1}} \right)^{r^*+1} \epsilon^{-\frac{1}{r^*}}.$$

(2,3) It the follows by a similar argument as in points (2, 3) of the proof of Proposition 8 that there is a constant $C_0$ such that with probability $1 - \delta$ for all $f \in \mathcal{F}$

$$\mathcal{L}(f) - \tilde{\mathcal{L}}_N(f) \le C_0 \left( \sum_{\ell=1}^{L} (\rho_{L:\ell+1} C_\ell R_\ell)^{\frac{1}{r^*+1}} \right)^{r^*+1} E_{r^*}(N) + c_0 \sqrt{\frac{2 \log 2/\delta}{N}}$$

$\qquad\square$

# D  Generalization at the Regularized Global Minimum

In this section, we first give the proof of Theorem 5 and then present detailed proofs of lemmas used in the proof. The lemmas are largely inspired by [5] and may be of independent interest.

 **D.1 Theorem 5 in Section 4.2**

**Theorem 11** (Theorem 5 in the main). *Given a true function $f^*_{L^*:1} = f^*_{L^*} \circ \cdots \circ f^*_1$ going through the dimensions $d^*_0, \ldots, d^*_{L^*}$, along with a continuous input distribution $\pi_0$ supported in $B(0, b_0)$, such that the distributions $\pi_\ell$ of $f^*_\ell(x)$ (for $x \sim \pi_0$) are continuous too and supported inside $B(0, b_\ell) \subset \mathbb{R}^{d^*_\ell}$. Further assume that there are differentiabilities $\nu_\ell$ and radii $R_\ell$ such that $\|f^*_\ell\|_{W^{\nu_\ell,2}(B(0,b_\ell))} \leq R_\ell$, and $\rho_\ell$ such that $Lip(f^*_\ell) \leq \rho_\ell$. For a infinite width AccNet with $L \geq L^*$ and constant width $d \geq d^*_1, \ldots, d^*_{L^*-1}$, we have for the ratios $\tilde{r}_\ell = \frac{\nu_\ell}{d^*_\ell + 3}$:*

- *At a global minimizer $\hat{f}_{L:1}$ of the regularized loss $f_1, \ldots, f_L \mapsto \tilde{\mathcal{L}}_N(f_{L:1}) + \lambda R(f_1, \ldots, f_L)$, we have $\mathcal{L}(\hat{f}_{L:1}) = \tilde{O}(N^{-\min\{\frac{1}{2}, \tilde{r}_1, \ldots, \tilde{r}_{L^*}\}})$.*

- *At a global minimizer $\hat{f}_{L:1}$ of the regularized loss $f_1, \ldots, f_L \mapsto \tilde{\mathcal{L}}_N(f_{L:1}) + \lambda \prod_{\ell=1}^L \|f_\ell\|_{F_1}$, we have $\mathcal{L}(\hat{f}_{L:1}) = \tilde{O}(N^{-\frac{1}{2} + \sum_{\ell=1}^{L^*} \max\{0, \tilde{r}_\ell - \frac{1}{2}\}})$.*

*Proof.* If $f^* = f^*_{L^*} \circ \cdots \circ f^*_1$ with $L^* \leq L$, intermediate dimensions $d^*_0, \ldots, d^*_{L^*}$, along with a continuous input distribution $\pi_0$ supported in $B(0, b_0)$, such that the distributions $\pi_\ell$ of $f^*_\ell(x)$ (for $x \sim \pi_0$) are continuous too and supported inside $B(0, b_\ell) \subset \mathbb{R}^{d^*_\ell}$. Further assume that there are differentiabilities $\nu^*_\ell$ and radii $R_\ell$ such that $\|f^*_\ell\|_{W^{\nu^*_\ell,2}(B(0,b_\ell))} \leq R_\ell$.

We first focus on the $L = L^*$ case and then extend to the $L > L^*$ case.

Each $f^*_\ell$ can be approximated by another function $\tilde{f}_\ell$ with bounded $F_1$-norm and Lipschitz constant. Actually if $2\nu^*_\ell \geq d^*_{\ell-1} + 3$ one can choose $\tilde{f}_\ell = f^*_\ell$ since $\|f^*_\ell\|_{F_1} \leq C_\ell R_\ell$ by Lemma 14, and by assumption $Lip(\tilde{f}_\ell) \leq \rho_\ell$. If $2\nu^*_\ell < d^*_{\ell-1} + 3$, then by Lemma 13 we know that there is a $\tilde{f}_\ell$ with $\left\|\tilde{f}_\ell\right\|_{F_1} \leq C_\ell R_\ell \epsilon_\ell^{-\frac{1}{2\tilde{r}_\ell} + 1}$ and $Lip(\tilde{f}_\ell) \leq C_\ell Lip(f^*_\ell) \leq C_\ell \rho_\ell$ and error

$$\left\|f^*_\ell - \tilde{f}_\ell\right\|_{L_2(\pi_{\ell-1})} \leq c_\ell \left\|f^* - \tilde{f}_\ell\right\|_{L_2(B(0,b_\ell))} \leq c_\ell \epsilon_\ell.$$

Therefore the composition $\tilde{f}_{L:1}$ satisfies

$$\begin{aligned}
\left\|f^*_{L:1} - \hat{f}_{L:1}\right\|_{L_2(\pi_{\ell-1})} &\leq \sum_{\ell=1}^L \left\|\tilde{f}_{L:\ell+1} \circ f^*_{\ell:1} - \tilde{f}_{L:\ell} \circ f^*_{\ell-1:1}\right\|_{L_2(\pi)} \\
&\leq \sum_{\ell=1}^L Lip(\tilde{f}_{L:\ell+1}) c_\ell \epsilon_\ell \\
&\leq \sum_{\ell=1}^L \rho_{L:\ell+1} C_{L:\ell+1} c_\ell \epsilon_\ell.
\end{aligned}$$

For any $L \geq L^*$, dimensions $d_\ell \geq d^*_\ell$ and widths $w_\ell \geq N$, we can build an AccNet that fits eactly $\tilde{f}_{L:1}$, by simply adding zero weights along the additional dimensions and widths, and by adding identity layers if $L > L^*$, since it is possible to represent the identity on $\mathbb{R}^d$ with a shallow network with $2d$ neurons and $F_1$-norm $2d$ (by having two neurons $e_i \sigma(e_i^T \cdot)$ and $-e_i \sigma(-e_i^T \cdot)$ for each basis $e_i$). Since the cost in parameter norm of representing the identity scales with the dimension, it is best to add those identity layers at the minimal dimension $\min\{d^*_0, \ldots, d^*_{L^*}\}$. We therefore end up with a AccNet with $L - L^*$ identity layers (with $F_1$ norm $2\min\{d^*_0, \ldots, d^*_{L^*}\}$) and $L^*$ layers that approximate each of the $f^*_\ell$ with a bounded $F_1$-norm function $\tilde{f}_\ell$.

Since $f^*_{L:1}$ has zero population loss, the population loss of the AccNet $\tilde{f}_{L:1}$ is bounded by $\rho \sum_{\ell=1}^L \rho_{L:\ell+1} C_{L:\ell+1} c_\ell \epsilon_\ell$. By McDiarmid's inequality, we know that with probability $1 - \delta$ over the sampling of the training set, the training loss is bounded by $\rho \sum_{\ell=1}^L \rho_{L:\ell+1} C_{L:\ell+1} c_\ell \epsilon_\ell + B\sqrt{\frac{2\log 2/\delta}{N}}$. (1) The global minimizer $\hat{f}_{L:1} = \hat{f}_L \circ \cdots \circ \hat{f}_1$ of the regularized loss (with the first regularization

term) is therefore bounded by

$$\rho \sum_{\ell=1}^{L} \rho_{L:\ell+1} C_{L:\ell+1} c_\ell \epsilon_\ell + B\sqrt{\frac{2\log {}^{2}/\delta}{N}}$$

$$+ \lambda\sqrt{2d}\left[\prod_{\ell=1}^{L^*} C_\ell \rho_\ell \sum_{\ell=1}^{L^*} \frac{1}{C_\ell \rho_\ell} \begin{cases} C_\ell R_\ell & 2\nu_\ell^* \geq d_{\ell-1}^* + 3 \\ C_\ell R_\ell \epsilon_\ell^{-\frac{1}{2\tilde{r}_\ell}+1} & 2\nu_\ell^* < d_{\ell-1}^* + 3 \end{cases} + 2(L-L^*)\min\{d_0^*, \ldots, d_{L^*}^*\}\right].$$

Taking $\epsilon_\ell = E_{\tilde{r}_{min}}(N)$ and $\lambda = N^{-\frac{1}{2}}\log N$, this is upper bounded by

$$\left[\rho\sum_{\ell=1}^{L}\rho_{L:\ell+1}C_{L:\ell+1}c_\ell + C\sqrt{2d}r\prod_{\ell=1}^{L^*}C_\ell\rho_\ell\sum_{\ell=1}^{L^*}\frac{R_\ell}{\rho_\ell} + 2(L-L^*)\min\{d_0^*,\ldots,d_{L^*}^*\}\right]E_{\tilde{r}_{min}}(N) + B\sqrt{\frac{2\log{}^2/\delta}{N}}.$$

which implies that at the globla minimizer of the regularized loss, the (unregularized) train loss is of order $E_{\tilde{r}_{min}}(N)$ and the complexity measure $R(\hat{f}_1,\ldots,\hat{f}_L)$ is of order $\frac{1}{N}E_{\tilde{r}_{min}}(N)$ which implies that the test error will be of order

$$\mathcal{L}(f) \leq \left[2\rho\sum_{\ell=1}^{L}\rho_{L:\ell+1}C_{L:\ell+1}c_\ell + 2C\sqrt{2d}r\prod_{\ell=1}^{L^*}C_\ell\rho_\ell\sum_{\ell=1}^{L^*}\frac{R_\ell}{\rho_\ell} + 2(L-L^*)\min\{d_0^*,\ldots,d_{L^*}^*\}\right]E_{\tilde{r}_{min}}(N)$$

$$+ (2B+c_0)\sqrt{\frac{2\log{}^2/\delta}{N}}.$$

(2) Let us now consider adding the closer to traditional $L_2$-regularization $\mathcal{L}_\lambda(f_{L:1}) = \mathcal{L}(f_{L:1}) + \lambda\prod_{\ell=1}^{L}\|f_\ell\|_{F_1}$. ,We see that the global minimizer $\hat{f}_{L:1}$ of the $L_2$-regularized loss is upper bounded by

$$\rho\sum_{\ell=1}^{L}\rho_{L:\ell+1}C_{L:\ell+1}c_\ell\epsilon_\ell + B\sqrt{\frac{2\log{}^2/\delta}{N}} + \lambda\left[\prod_{\ell=1}^{L^*}\begin{cases}C_\ell R_\ell & 2\nu_\ell^* \geq d_{\ell-1}^*+3 \\ C_\ell R_\ell\epsilon_\ell^{-\frac{1}{2\tilde{r}_\ell}+1} & 2\nu_\ell^* < d_{\ell-1}^*+3\end{cases}\right](2\min\{d_0^*,\ldots,d_{L^*}^*\})^{(L-L^*)}.$$

Which for $\epsilon_\ell = E_{\tilde{r}_{min}}(N)$ and $\lambda = N^{-\frac{1}{N}}$ is upper bounded by

$$\rho\sum_{\ell=1}^{L}\rho_{L:\ell+1}C_{L:\ell+1}c_\ell E_{\tilde{r}_{min}}(N) + B\sqrt{\frac{2\log{}^2/\delta}{N}} + N^{-\frac{1}{2}}\left[\prod_{\ell=1}^{L^*}C_\ell R_\ell\sqrt{N}E_{\tilde{r}_{min}}(N)\right](2\min\{d_0^*,\ldots,d_{L^*}^*\})^{(L-L^*)}.$$

Which implies that both the train error is of order $N^{-\frac{1}{2}}\prod_{\ell=1}^{L^*}\sqrt{N}E_{\tilde{r}_{min}}(N)$ and the product of the $F_1$-norms is of order $\prod_{\ell=1}^{L^*}\sqrt{N}E_{\tilde{r}_{min}}(N)$.

Now note that the product of the $F_1$-norms bounds the complexity measure up to a constant since $Lip(f) \leq \|f\|_{F_1}$

$$R(f_1,\ldots,f_L) = r\prod_{\ell=1}^{L}Lip(f_\ell)\sum_{\ell=1}^{L}\frac{\|f_\ell\|_{F_1}}{Lip(f_\ell)}\sqrt{d_{\ell-1}+d_\ell} \leq L\sqrt{2d}\prod_{\ell=1}^{L}\|f\|_{F_1}.$$

And since at the global minimum the product of the $F_1$-norms is of order $\prod_{\ell=1}^{L^*}\sqrt{N}E_{\tilde{r}_{min}}(N)$ the test error will of order $\left(\prod_{\ell=1}^{L^*}\sqrt{N}E_{\tilde{r}_\ell}(N)\right)\frac{\log N}{\sqrt{N}}$.

Note that if there is at a most one $\ell$ where $\tilde{r}_\ell > \frac{1}{2}$ then the rate is up to log term the same as $E_{\tilde{r}_{min}}(N)$. $\qquad\square$

## D.2 Lemmas on approximating Sobolev functions

Now we present the lemmas used in this proof above that concern the approximation errors and Lipschitz constants of Sobolev functions and compositions of them. We will bound the $F_2$-norm and note that the $F_2$-norm is larger than the $F_1$-norm, cf. [5, Section 3.1].

**Lemma 12** (Approximation for Sobolev function with bounded error and Lipschitz constant).
*Suppose $g : \mathbb{S}_d \to \mathbb{R}$ is an even function with bounded Sobolev norm $\|g\|^2_{W^{\nu,2}} \le R$ with $2\nu \le d + 2$, with inputs on the unit $d$-dimensional sphere. Then for every $\epsilon > 0$, there is $\hat{g} \in \mathcal{G}_2$ with small approximation error $\|g - \hat{g}\|_{L_2(\mathbb{S}_d)} = C(d, \nu, R)\epsilon$, bounded Lipschitzness $\mathrm{Lip}(\hat{g}) \le C'(d)\mathrm{Lip}(g)$, and bounded norm*

$$\|\hat{g}\|_{F_2} \le C''(d, \nu, R)\epsilon^{-\frac{d+3-2\nu}{2\nu}}.$$

*Proof.* Given our assumptions on the target function $g$, we may decompose $g(x) = \sum_{k=0}^{\infty} g_k(x)$ along the basis of spherical harmonics with $g_0(x) = \int_{\mathbb{S}_d} g(y)\mathrm{d}\tau_d(y)$ being the mean of $g(x)$ over the uniform distribution $\tau_d$ over $\mathbb{S}_d$. The $k$-th component can be written as

$$g_k(x) = N(d, k) \int_{\mathbb{S}_d} g(y)P_k(x^T y)\mathrm{d}\tau_d(y)$$

with $N(d, k) = \frac{2k+d-1}{k}\binom{k+d-2}{d-1}$ and a Gegenbauer polynomial of degree $k$ and dimension $d + 1$:

$$P_k(t) = (-1/2)^k \frac{\Gamma(d/2)}{\Gamma(k + d/2)}(1 - t^2)^{(2-d)/2}\frac{d^k}{dt^k}(1 - t^2)^{k+(d-2)/2},$$

known as Rodrigues' formula. Given the assumption that the Sobolev norm $\|g\|^2_{W^{\nu,2}}$ is upper bounded, we have $\|f\|^2_{L_2(\mathbb{S}_d)} \le C_0(d, \nu)R$ for $f = \Delta^{\nu/2}g$ where $\Delta$ is the Laplacian on $\mathbb{S}_d$ [18, 5]. Note that $g_k$ are eigenfunctions of the Laplacian with eigenvalues $k(k + d - 1)$ [4], thus

$$\|g_k\|^2_{L_2(\mathbb{S}_d)} = \|f_k\|^2_{L_2(\mathbb{S}_d)}(k(k + d - 1))^{-\nu} \le \|f_k\|^2_{L_2(\mathbb{S}_d)}k^{-2\nu} \le C_1(d, \nu, R)k^{-2\nu-1} \quad (1)$$

where the last inequality holds because $\|f\|^2_{L_2(\mathbb{S}_d)} = \sum_{k\ge 0} \|f_k\|^2_{L_2(\mathbb{S}_d)}$ converges. Note using the Hecke-Funk formula, we can also write $g_k$ as scaled $p_k$ for the underlying density $p$ of the $F_1$ and $F_2$-norms:

$$g_k(x) = \lambda_k p_k(x)$$

where $\lambda_k = \frac{\omega_{d-1}}{\omega_d}\int_{-1}^{1}\sigma(t)P_k(t)(1 - t^2)^{(d-2)/2}\mathrm{d}t = \Omega(k^{-(d+3)/2})$ [5, Appendix D.2] and $\omega_d$ denotes the surface area of $\mathbb{S}_d$. Then by definition of $\|\cdot\|_{F_2}$, for some probability density $p$,

$$\|g\|^2_{F_2} = \int_{\mathbb{S}_d} |p|^2\mathrm{d}\tau(v) = \|p\|^2_{L_2(\mathbb{S}_d)} = \sum_{0\le k}\|p_k\|^2_{L_2(\mathbb{S}_d)} = \sum_{0\le k}\lambda_k^{-2}\|g_k\|^2_{L_2(\mathbb{S}_d)}.$$

Now to approximate $g$, consider function $\hat{g}$ defined by truncating the "high frequencies" of $g$, i.e. setting $\hat{g}_k = \mathbb{1}[k \le m]g_k$ for some $m > 0$ we specify later. Then we can bound the norm with

$$\|\hat{g}\|^2_{F_2} = \sum_{0\le k:\lambda_k \ne 0}\lambda_k^{-2}\|\hat{g}_k\|^2_{L_2(\mathbb{S}_d)} = \sum_{\substack{0\le k\le m \\ \lambda_k \ne 0}}\lambda_k^{-2}\|g_k\|^2_{L_2(\mathbb{S}_d)}$$

$$\overset{(a)}{\le} C_2(d, \nu, R)\sum_{0\le k\le m}k^{d+2-2\nu}$$

$$\overset{(b)}{\le} C_3(d, \nu, R)m^{d+3-2\nu}$$

where (a) uses Eq 1 and $\lambda_k = \Omega(k^{-(d+3)/2})$; (b) approximates by integral.

To bound the approximation error,

$$\|g - \hat{g}\|^2_{L_2(\mathbb{S}_d)} = \left\|\sum_{k>m} g_k\right\|^2_{L_2(\mathbb{S}_d)} \le \sum_{k>m}\|g_k\|^2_{L_2(\mathbb{S}_d)}$$

$$\le C_4(d, \nu, R)\sum_{k>m}k^{-2\nu-1}$$

$$\le C_5(d, \nu, R)m^{-2\nu} \quad \text{by integral approximation.}$$

Finally, choosing $m = \epsilon^{-\frac{1}{\nu}}$, we obtain $\|g - \hat{g}\|_{L_2(\mathbb{S}_d)} \le C(d, \nu, R)\epsilon$ and

$$\|\hat{g}\|_{F_2} \le C'(d, \nu, R)\epsilon^{-\frac{d+3-2\nu}{2\nu}}.$$

Then it remains to bound $\mathrm{Lip}(\hat{g})$ for our constructed approximation. By construction and by [13, Theorem 2.1.3], we have $\hat{g} = g * h$ with now

$$h(t) = \sum_{k=0}^{m} h_k P_k(t), \quad t \in [-1, 1]$$

by orthogonality of the Gegenbauer polynomial $P_k$'s and the convolution is defined as

$$(g * h)(x) := \frac{1}{\omega_d} \int_{\mathbb{S}_d} g(y) h(\langle x, y \rangle) \mathrm{d}y.$$

The coefficients for $0 \le k \le m$ given by [13, Theorem 2.1.3] are

$$h_k \overset{(a)}{=} \frac{\omega_{d+1}}{\omega_d} \frac{\Gamma(d-1)}{\Gamma(d-1+k)} P_k(1) \frac{k!(k+(d-1)/2)\Gamma((d-1)/2)^2}{\pi 2^{2-d}\Gamma(d-1+k)} \overset{(b)}{=} O\left(\frac{k}{\Gamma(d-1+k)}\right)$$

where (a) follows from the (inverse of) weighted $L_2$ norm of $P_k$; (b) plugs in the unit constant $P_k(1) = \frac{\Gamma(k+d-1)}{\Gamma(d-1)k!}$ and suppresses the dependence on $d$. Note that the constant factor $\frac{\Gamma(d-1)}{\Gamma(d-1+k)}$ comes from the difference in the definitions of the Gegenbauer polynomials here and in [13]. Then we can bound

$$\|\nabla \hat{g}(x)\|_{op} \le \int_{\mathbb{S}_d} \|\nabla g(y)\|_{op} |h(\langle x, y \rangle)| \mathrm{d}y$$

$$\le \mathrm{Lip}(g) \int_{\mathbb{S}_d} |h(\langle x, y \rangle)| \mathrm{d}y$$

$$\le \sqrt{\omega_d} \mathrm{Lip}(g) \left( \int_{\mathbb{S}_d} h(\langle x, y \rangle)^2 \mathrm{d}y \right)^{1/2} \qquad \text{by Cauchy-Schwartz}$$

$$= \sqrt{\omega_d} \mathrm{Lip}(g) \left( \sum_{k,j=0}^{m} \int_{\mathbb{S}_d} h_k h_j P_k(\langle x, y \rangle) P_j(\langle x, y \rangle) \mathrm{d}y \right)^{1/2}$$

$$= \sqrt{\omega_d} \mathrm{Lip}(g) \left( \sum_{k,j=0}^{m} \int_{-1}^{1} h_k h_j P_k(t) P_j(t) (1-t^2)^{\frac{d-2}{2}} \mathrm{d}t \right)^{1/2} \qquad \text{by [13, Eq A.5.1]}$$

$$= \sqrt{\omega_d} \mathrm{Lip}(g) \left( \sum_{k=0}^{m} h_k^2 \int_{-1}^{1} P_k(t)^2 (1-t^2)^{\frac{d-2}{2}} \mathrm{d}t \right)^{1/2} \qquad \text{by orthogonality of } P_k\text{'s w.r.t. this measure}$$

$$= \sqrt{\omega_d} \mathrm{Lip}(g) \left( \sum_{k=0}^{m} h_k^2 \frac{\pi 2^{2-d}\Gamma(d-1+k)}{k!(k+(d-1)/2)\Gamma((d-1)/2)^2} \right)^{1/2}$$

$$= \sqrt{\omega_d} \mathrm{Lip}(g) \left( O(1) + \sum_{k=1}^{m} O\left( \frac{k}{\Gamma(d-1+k)k!} \right) \right)^{1/2}$$

$$= \sqrt{\omega_d} \mathrm{Lip}(g) C(d)$$

for some constant $C(d)$ that only depends on $d$. Hence $\mathrm{Lip}(\hat{g}) = C'(d)\mathrm{Lip}(g)$. $\qquad \square$

The next lemma adapts Lemma 12 to inputs on balls instead of spheres following the construction in [5, Proposition 5].

**Lemma 13.** *Suppose $f : B(0, b) \to \mathbb{R}$ has bounded Sobolev norm $\|f\|_{W^{\nu,2}}^2 \le R$ with $\nu \le (d+2)/2$ even, where $B(0, b) = \{x \in \mathbb{R}^d : \|x\|_2 \le b\}$ is the radius-$b$ ball. Then for every $\epsilon > 0$ there exists $f_\epsilon \in \mathcal{F}_2$ such that $\|f - f_\epsilon\|_{L_2(B(0,b))} = C(d, \nu, b, R)\epsilon$, $\mathrm{Lip}(f_\epsilon) \le C'(b, d)\mathrm{Lip}(f)$, and*

$$\|f_\epsilon\|_{F_2} \le C''(d, \nu, b, R)\epsilon^{-\frac{d+3-2\nu}{2\nu}}$$

*Proof.* Define $g(z, a) = f\left(\frac{2bz}{a}\right) a$ on $(z, a) \in \mathbb{S}_d$ with $z \in \mathbb{R}^d$ and $\frac{1}{\sqrt{2}} \le a \in \mathbb{R}$. One may verify that unit-norm $(z, a)$ with $a \ge \frac{1}{\sqrt{2}}$ is sufficient to cover $B(0, b)$ by setting $x = \frac{bz}{a}$ and

714  solve for $(z, a)$. Then we have bounded $\|g\|_{W^{\nu,2}}^2 \leq b^\nu R$ and may apply Lemma 12 to get $\hat{g}$ with
715  $\|g - \hat{g}\|_{L_2(\mathbb{S}_d)} \leq C(d, \nu, b, R)\epsilon$. Letting $f_\epsilon(x) = \hat{g}\left(\frac{ax}{b}, a\right) a^{-1}$ for the corresponding $\left(\frac{ax}{b}, a\right) \in \mathbb{S}_d$
716  gives the desired upper bounds.  $\square$

717  **Lemma 14.** *Suppose $f : B(0, b) \to \mathbb{R}$ has bounded Sobolev norm $\|f\|_{W^{\nu,2}}^2 \leq R$ with $\nu \geq (d+3)/2$*
718  *even. Then $f \in \mathcal{F}_2$ and $\|f\|_{F_2} \leq C(d, \nu)b^\nu R$.*

719  *In particular, $W^{\nu,2} \subseteq \mathcal{F}_2$ for $\nu \geq (d+3)/2$ even.*

720  *Proof.* This lemma reproduces [5, Proposition 5] to functions with bounded Sobolev $L_2$ norm instead
721  of $L_\infty$ norm. The proof follows that of Lemma 12 and Lemma 13 and noticing that by Eq 1,

$$\begin{aligned}
\|g\|_{F_2}^2 &= \sum_{0 \leq k : \lambda_k \neq 0} \lambda_k^{-2} \|g_k\|_{L_2(\mathbb{S}_d)}^2 \\
&\leq \sum_{0 \leq k} k^{d+3-2\nu} \|(\Delta^{\nu/2} g)_k\|_{L_2(\mathbb{S}_d)}^2 \\
&\leq \|\Delta^{\nu/2} g\|_{L_2(\mathbb{S}_d)}^2 \\
&\leq C_1(d, \nu) \|g\|_{W^{\nu,2}}^2 \\
&\leq C_1(d, \nu) R.
\end{aligned}$$

722  $\square$

723  Finally, we remark that the above lemmas extend straightforward to functions $f : B(0, b) \to \mathbb{R}^{d'}$
724  with multi-dimensional outputs, where the constants then depend on the output dimension $d'$ too.

### D.3 Lemma on approximating compositions of Sobolev functions

726  With the lemmas given above and the fact that the $F_2$-norm upper bounds the $F_1$-norm, we can find
727  infinite-width DNN approximations for compositions of Sobolev functions, which is also pointed out
728  in the proof of Theorem 5.

729  **Lemma 15.** *Assume the target function $f : \Omega \to \mathbb{R}^{d_{out}}$, with $\Omega \subseteq B(0, b) \subseteq \mathbb{R}^{d_{in}}$, satisfies:*

730  • *$f = g_k \circ \cdots \circ g_1$ a composition of $k$ Sobolev functions $g_i : \mathbb{R}^{d_i} \to \mathbb{R}^{d_{i+1}}$ with bounded*
731  *norms $\|g_i\|_{W^{\nu_i,2}}^2 \leq R$ for $i = 1, \ldots, k$, with $d_1 = d_{in}$;*

732  • *$f$ is Lipschitz, i.e. $\text{Lip}(g_i) < \infty$ for $i = 1, \ldots, k$.*

733  *If $\nu_i \leq (d_i + 2)/2$ for any $i$, i.e. less smooth than needed, for depth $L \geq k$ and any $\epsilon > 0$, there is an*
734  *infinite-width DNN $\tilde{f}$ such that*

735  • *$\text{Lip}(\tilde{f}) \leq C_1 \prod_{i=1}^k \text{Lip}(g_i)$;*

736  • *$\|\tilde{f} - f\|_{L_2} \leq C_2 \epsilon$;*

737  *the constants $C_1$ depends on all of the input dimensions $d_i$ (to $g_i$) and $d_{out}$, and $C_2$ depends on*
738  *$d_i, d_{out}, \nu_i, b, R, k$, and $\text{Lip}(g_i)$ for all $i$.*

739  *If otherwise $\nu_i \geq (d_i + 3)/2$ for all $i$, we can have $\tilde{f} = f$ where each layer has a parameter norm*
740  *bounded by $C_3 R$, with $C_3$ depending on $d_i, d_{out}, \nu_i$, and $b$.*

741  *Proof.* Note that by Lipschitzness,

$$(g_i \circ \cdots \circ g_1)(\Omega) \subseteq B\left(0, b \prod_{j=1}^i \text{Lip}(g_j)\right),$$

742  i.e. the pre-image of each component lies in a ball. By Lemma 12, for each $g_i$, if $\nu_i \leq (d_i + 2)/2$,
743  we have an approximation $\hat{g}_i$ on a slightly larger ball $b'_i = b \prod_{j=1}^{i-1} C''(d_j, d_{j+1})\text{Lip}(g_j)$ such that

744 • $\|g_i - \hat{g}_i\|_{L_2} \leq C(d_i, d_{i+1}, \nu_i, b'_i, R)\epsilon$;

745 • $\|\hat{g}_i\|_{F_2} \leq C'(d_i, d_{i+1}, \nu_i, b'_i, R)\epsilon^{\frac{d_i+3-2\nu_i}{2\nu_i}}$;

746 • $\text{Lip}(\hat{g}_i) \leq C''(d_i, d_{i+1})\text{Lip}(g_i)$;

747 where $d_i$ is the input dimension of $g_i$. Write the constants as $C_i$, $C'_i$, and $C''_i$ for notation simplicity.
748 Note that the Lipschitzness of the approximations $\hat{g}_i$'s guarantees that, when they are composed,
749 $(\hat{g}_{i-1} \circ \cdots \circ \hat{g}_1)(\Omega)$ lies in a ball of radius $b'_i = b\prod_{j=1}^{i-1} C''_j \text{Lip}(g_j)$, hence the approximation error
750 remains bounded while propagating. While each $\hat{g}_i$ is a (infinite-width) layer, for the other $L - k$
751 layers, we may have identity layers[5].

752 Let $\tilde{f}$ be the composed DNN of these layers. Then we have

$$\text{Lip}(\tilde{f}) \leq \prod_{i=1}^{k} C''_i \text{Lip}(g_i) = C''(d_1, \ldots, d_k, d_{out})\prod_{i=1}^{k} \text{Lip}(g_i)$$

753 and approximation error

$$\|\tilde{f} - f\|_{L_2} \leq \sum_{i=1}^{k} C_i\epsilon \prod_{j>i} C''_j \text{Lip}(g_j) = O(\epsilon)$$

754 where the last equality suppresses the dependence on $d_i, d_{out}, \nu_i, b, R, k$, and $\text{Lip}(g_i)$ for $i =$
755 $1, \ldots, k$.

756 In particular, by Lemma 14, if $\nu_i \geq (d_i + 3)/2$ for any $i = 1, \ldots, k$, we can take $\hat{g}_i = g_i$. If this
757 holds for all $i$, then we can have $\tilde{f} = f$ while each layer has a $F_2$-norm bounded by $O(R)$. $\qquad\square$

# E   Technical results

759 Here we show a number of technical results regarding the covering number.

760 First, here is a bound for the covering number of Ellipsoids, which is a simple reformulation of
761 Theorem 2 of [17]:

762 **Theorem 16.** *The $d$-dimensional ellipsoid $E = \{x : x^T K^{-1}x \leq 1\}$ with radii $\sqrt{\lambda_i}$ for $\lambda_i$ the $i$-th*
763 *eigenvalue of $K$ satisfies $\log \mathcal{N}_2(E, \epsilon) = M_\epsilon(1 + o(1))$ for*

$$M_\epsilon = \sum_{i:\sqrt{\lambda_i} \geq \epsilon} \log \frac{\sqrt{\lambda_i}}{\epsilon}$$

764 *if one has $\log \frac{\sqrt{\lambda_1}}{\epsilon} = o\left(\frac{M_\epsilon^2}{k_\epsilon \log d}\right)$ for $k_\epsilon = \left|\{i : \sqrt{\lambda_i} \geq \epsilon\}\right|$*

765 For our purpose, we will want to cover a unit ball $B = \{w : \|w\| \leq 1\}$ w.r.t. to a non-isotropic norm
766 $\|w\|_K^2 = w^T K w$, but this is equivalent to covering $E$ with an isotropic norm:

767 **Corollary 17.** *The covering number of the ball $B = \{w : \|w\| \leq 1\}$ w.r.t. the norm $\|w\|_K^2 = w^T K w$*
768 *satisfies $\log \mathcal{N}(B, \|\cdot\|_K, \epsilon) = M_\epsilon(1 + o(1))$ for the same $M_\epsilon$ as in Theorem 16 and under the same*
769 *condition.*

770 *Furthermore, $\log \mathcal{N}(B, \|\cdot\|_K, \epsilon) \leq \frac{\text{Tr}K}{2\epsilon^2}(1 + o(1))$ as long as $\log d = o\left(\frac{\sqrt{\text{Tr}K}}{\epsilon}\left(\log \frac{\sqrt{\text{Tr}K}}{\epsilon}\right)^{-1}\right)$.*

771 *Proof.* If $\tilde{E}$ is an $\epsilon$-covering of $E$ w.r.t. to the $L_2$-norm, then $\tilde{B} = K^{-\frac{1}{2}}\tilde{E}$ is an $\epsilon$-covering of $B$
772 w.r.t. the norm $\|\cdot\|_K$, because if $w \in B$, then $\sqrt{K}w \in E$ and so there is an $\tilde{x} \in \tilde{E}$ such that
773 $\left\|x - \sqrt{K}w\right\| \leq \epsilon$, but then $\tilde{w} = \sqrt{K}^{-1}x$ covers $w$ since $\|\tilde{w} - w\|_K = \left\|x - \sqrt{K}w\right\|_K \leq \epsilon$.

---

[5]Since the domain is always bounded here, one can let the bias translate the domain to the first quadrant and let the weight be the identity matrix, cf. the construction in [45, Proposition B.1.3].

774   Since $\lambda_i \leq \frac{\text{Tr}K}{i}$, we have $K \leq \bar{K}$ for $\bar{K}$ the matrix obtained by replacing the $i$-th eigenvalue $\lambda_i$ of
775   $K$ by $\frac{\text{Tr}K}{i}$, and therefore $\mathcal{N}(B, \|\cdot\|_K, \epsilon) \leq \mathcal{N}(B, \|\cdot\|_{\bar{K}}, \epsilon)$ since $\|\cdot\|_K \leq \|\cdot\|_{\bar{K}}$. We now have the
776   a[proximation $\log \mathcal{N}(B, \|\cdot\|_{\bar{K}}, \epsilon) = \bar{M}_\epsilon(1 + o(1))$ for

$$\bar{M}_\epsilon = \sum_{i=1}^{\bar{k}_\epsilon} \log \frac{\sqrt{\text{Tr}K}}{\sqrt{i}\epsilon}$$

$$\bar{k}_\epsilon = \left\lfloor \frac{\text{Tr}K}{\epsilon^2} \right\rfloor.$$

777   We now have the simplification

$$\bar{M}_\epsilon = \sum_{i=1}^{k_\epsilon} \log \frac{\sqrt{\text{Tr}K}}{\sqrt{i}\epsilon} = \frac{1}{2}\sum_{i=1}^{\bar{k}_\epsilon}\log\frac{\bar{k}_\epsilon}{i} = \frac{\bar{k}_\epsilon}{2}\left(\int_0^1 \log\frac{1}{x}dx + o(1)\right) = \frac{\bar{k}_\epsilon}{2}(1 + o(1))$$

778   where the $o(1)$ term vanishes as $\epsilon \searrow 0$. Furthermore, this allows us to check that as long as
779   $\log d = o\left(\frac{\sqrt{\text{Tr}K}}{4\epsilon \log \frac{\sqrt{\text{Tr}K}}{\epsilon}}\right)$, the condition is satisfied

$$\log \frac{\sqrt{\text{Tr}K}}{\epsilon} = o\left(\frac{\bar{k}_\epsilon}{4\log d}\right) = o\left(\frac{\bar{M}_\epsilon^2}{\bar{k}_\epsilon \log d}\right).$$

780   □

781   Second we prove how to obtain the covering number of the convex hull of a function set $\mathcal{F}$:

782   **Theorem 18.** *Let $\mathcal{F}$ be a set of $B$-uniformly bounded functions, then for all $\epsilon_K = B2^{-K}$*

$$\sqrt{\log \mathcal{N}_2(\text{Conv}\mathcal{F}, 2\epsilon_K)} \leq \sqrt{18}\sum_{k=1}^{K} 2^{K-k}\sqrt{\log \mathcal{N}_2(\mathcal{F}, B2^{-k})}.$$

783   *Proof.* Define $\epsilon_k = B2^{-k}$ and the corresponding $\epsilon_k$-coverings $\tilde{\mathcal{F}}_k$ (w.r.t. some measure $\pi$). For any
784   $f$, we write $\tilde{f}_k[f]$ for the function $\tilde{f}_k[f] \in \tilde{\mathcal{F}}_k$ that covers $f$. Then for any functions $f$ in $\text{Conv}\mathcal{F}$, we
785   have

$$f = \sum_{i=1}^{m}\beta_i f_i = \sum_{i=1}^{m}\beta_i\left(f_i - \tilde{f}_K[f_i]\right) + \sum_{k=1}^{K}\sum_{i=1}^{m}\beta_i\left(\tilde{f}_k[f_i] - \tilde{f}_{k-1}[f_i]\right) + \tilde{f}_0[f_i].$$

786   We may assume that $\tilde{f}_0[f_i] = 0$ since the zero function $\epsilon_0$-covers the whole $\mathcal{F}$ since $\epsilon_0 = B$.

787   We will now use the probabilistic method to show that the sums $\sum_{i=1}^{m}\beta_i\left(\tilde{f}_k[f_i] - \tilde{f}_{k-1}[f_i]\right)$
788   can be approximated by finite averages. Consider the random functions $\tilde{g}_1^{(k)}, \ldots, \tilde{g}_{m_k}^{(k)}$
789   sampled iid with $\mathbb{P}\left[\tilde{g}_j^{(k)}\right] = \left(\tilde{f}_k[f_i] - \tilde{f}_{k-1}[f_i]\right)$ with probability $\beta_i$. We have $\mathbb{E}[\tilde{g}_j^{(k)}] =$
790   $\sum_{i=1}^{m}\beta_i\left(\tilde{f}_k[f_i] - \tilde{f}_{k-1}[f_i]\right)$ and

$$\mathbb{E}\left\|\sum_{k=1}^{K}\frac{1}{m_k}\sum_{j=1}^{m_k}\tilde{g}_j^{(k)} - \sum_{k=1}^{K}\sum_{i=1}^{m}\beta_i\left(\tilde{f}_k[f_i] - \tilde{f}_{k-1}[f_i]\right)\right\|_{L_p(\pi)}^{p} \leq \sum_{k=1}^{K}\frac{1}{m_k^p}\sum_{j=1}^{m_k}\mathbb{E}\left\|\tilde{g}_j^{(k)}\right\|_{L_p(\pi)}^{p}$$

$$= \sum_{k=1}^{K}\frac{1}{m_k}\sum_{i=1}^{m}\beta_i\left\|\tilde{f}_k[f_i] - \tilde{f}_{k-1}[f_i]\right\|_{L_p(\pi)}^{p}$$

$$\leq \sum_{k=1}^{K}\frac{3^2\epsilon_k^2}{m_k}.$$

Thus if we take $m_k = \frac{1}{a_k}(\frac{3\epsilon_k}{\epsilon_K})^2$ with $\sum a_k = 1$ we know that there must exist a choice of $\tilde{g}_j^{(k)}$'s such that

$$\left\| \sum_{k=1}^{K} \frac{1}{m_k} \sum_{j=1}^{m_k} \tilde{g}_j^{(k)} - \sum_{k=1}^{K} \sum_{i=1}^{m} \beta_i \left( \tilde{f}_k[f_i] - \tilde{f}_{k-1}[f_i] \right) \right\|_{L_p(\pi)} \leq \epsilon_K.$$

This implies that finite the set $\tilde{\mathcal{C}} = \left\{ \sum_{k=1}^{K} \frac{1}{m_k} \sum_{j=1}^{m_k} \tilde{g}_j^{(k)} : \tilde{g}_j^{(k)} \in \tilde{\mathcal{F}}_k - \tilde{\mathcal{F}}_{k-1} \right\}$ is an $2\epsilon_K$ covering of $\mathcal{C} = \text{Conv}\mathcal{F}$, since we know that for all $f = \sum_{i=1}^{m} \beta_i f_i$ there are $\tilde{g}_j^{(k)}$ such that

$$\left\| \sum_{k=1}^{K} \frac{1}{m_k} \sum_{j=1}^{m_k} \tilde{g}_j^{(k)} - \sum_{i=1}^{m} \beta_i f_i \right\|_{L_p(\pi)} \leq \left\| \sum_{i=1}^{m} \beta_i \left( f_i - \tilde{f}_K[f_i] \right) \right\|_{L_p(\pi)}$$

$$+ \sum_{k=1}^{K} \left\| \frac{1}{m_k} \sum_{j=1}^{m_k} \tilde{g}_j^{(k)} - \sum_{i=1}^{m} \beta_i \left( \tilde{f}_k[f_i] - \tilde{f}_{k-1}[f_i] \right) \right\|_{L_p(\pi)}$$

$$\leq 2\epsilon_K.$$

Since $\left| \tilde{\mathcal{C}} \right| = \prod_{k=1}^{K} \left| \tilde{\mathcal{F}}_k \right|^{m_k} \left| \tilde{\mathcal{F}}_{k-1} \right|^{m_k}$, we have

$$\log \mathcal{N}_p(\mathcal{C}, 2\epsilon_K) \leq \sum_{k=1}^{K} \frac{1}{a_k} (\frac{3\epsilon_k}{\epsilon_K})^2 \left( \log \mathcal{N}_p(\mathcal{F}, \epsilon_k) + \log \mathcal{N}_p(\mathcal{F}, \epsilon_{k-1}) \right)$$

$$\leq 18 \sum_{k=1}^{K} \frac{1}{a_k} 2^{2(K-k)} \log \mathcal{N}_2(\mathcal{F}, \epsilon_k).$$

This is minimized for the choice

$$a_k = \frac{2^{(K-k)} \sqrt{\log \mathcal{N}_2(\mathcal{F}, \epsilon_k)}}{\sum 2^{(K-k)} \sqrt{\log \mathcal{N}_2(\mathcal{F}, \epsilon_k)}},$$

which yields the bound

$$\sqrt{\log \mathcal{N}_p(\mathcal{C}, 2\epsilon_K)} \leq \sqrt{18} \sum_{k=1}^{K} 2^{K-k} \sqrt{\log \mathcal{N}_2(\mathcal{F}, \epsilon_k)}$$

$\square$

