# OpenReview forum: "How DNNs break the Curse of Dimensionality: Compositionality and Symmetry Learning"
_NeurIPS.cc/2024/Conference — Submitted to NeurIPS 2024_

### Official Review · Reviewer_vTqY · 2024-07-06

**Soundness:** 3
**Presentation:** 4
**Contribution:** 3
**Rating:** 5
**Confidence:** 3

**Summary:**

This paper introduces Accordion Networks (AccNets), a novel neural network structure composed of multiple shallow networks. The authors propose a generalization bound for AccNets that leverages the F1-norms and Lipschitz constants of the subnetworks, demonstrating that these networks can break the curse of dimensionality by efficiently learning compositions of Sobolev functions. The paper also provides theoretical insights and empirical validation, showcasing the superior performance of AccNets in learning complex compositional tasks compared to shallow networks and kernel methods.

**Strengths:**

The introduction of Accordion Networks (AccNets) as a novel neural network structure is a creative and original contribution. The paper provides a thorough theoretical analysis supported by empirical evidence, ensuring the soundness of its claims. The ability of AccNets to break the curse of dimensionality by learning compositional functions efficiently addresses a fundamental challenge in high-dimensional learning tasks.

**Weaknesses:**

1. The practical implementation of the proposed regularization methods might be challenging, particularly the first one requiring infinite width.

2. The paper mentions the difficulty in optimizing Lipschitz constants, which could be a limitation in practical applications.

3.  Additional experiments on more diverse real-world datasets could further demonstrate the robustness and generalizability of AccNets.

4. Although the author has discussed the differences between DNN and AccNet, there is still not enough information for me to be sure in which settings to use AccNet and in which settings to use DNN. More clear differences and applicable conditions, especially the shortcomings of each need to be pointed out.

**Questions:**

Can the authors provide more details on the computational complexity of training Accordion Networks compared to traditional DNNs?

How sensitive are the generalization bounds to the choice of hyperparameters, particularly the Lipschitz constants and F1-norms?

Are there any specific types of tasks or datasets where Accordion Networks might not perform as well as traditional methods?

**Limitations:**

See weaknesses.

---

### Official Review · Reviewer_eK3L · 2024-07-09

**Soundness:** 4
**Presentation:** 4
**Contribution:** 4
**Rating:** 7
**Confidence:** 3

**Summary:**

The authors present a generalization bound for deep neural networks that describes how depth enables models to learn functions that are compositions of Sobolev functions. To do this, they both prove a generalization bound for compositions of accordion networks (densely connected networks with a low-rank weight structure) and for compositions of Sobolev functions. They then present a sample efficiency result for different kinds of regularization on accordion networks.

**Strengths:**

I really liked this paper and would like to see it accepted to NeurIPS. It addresses an important question: how does depth change generalization bounds for deep neural networks? To my knowledge, not many papers so far have addressed this question and I found the findings presented here very interesting and well embedded within prior methodology.

I also found the paper very well written. I found it easy to follow along despite the highly technical nature of the results (note that I did not check the proofs in particular detail). I especially appreciated the remarks explaining different potential extensions and limitations.

Finally, the theory appears to be able to explain certain empirical phenomena (in networks trained under realistic paradigms) at least qualitatively (though note that I had a few questions I will mention under weaknesses and questions). This indicates to me that it is a promising way for thinking about generalization in deep neural networks.

**Weaknesses:**

1. I would like to see a more thorough comparison with shallow networks and generalization bounds, as this comparison is a central argument for the usefulness of the presented theory. While it is clear how the findings for the shallow network are a special case of the findings on the deep networks (as presented in Thm. 1), it remains a bit unclear to me how the theory can explain improved generalization in deep compared to shallow networks. The authors certainly present different several pieces of evidence on this: both Fig. 1 and Fig. 3 demonstrate that shallow networks exhibit worse scaling. I also appreciated the theoretical explanation of a particular contrast in l. 256-261. However, I think it would be really useful to provide a general theoretical explanation for this difference and test it empirically: would it be possible to extend the theoretical comparison in l. 256-261 to the general experimental setup studied in the figures --- and if so, would this theoretical comparison predict the conditions under which deep networks have the strongest advantages over shallow networks (or perhaps the conditions under which they don't perform that much better)? Not only would this serve as a useful validation of the theory, I think it would also provide a more extensive intuition for the authors' findings.

2. I appreciated the fact that the authors compare their findings with related work wherever this becomes relevant. However, I think a (potentially brief) section comparing the results here to other theoretical investigations of depth in deep networks (perhaps using different approaches) would be useful.

3. The linked codebase does not contain the notebooks indicated in the README as far as I can tell and therefore currently can't be used to directly reproduce the findings.

4. I believe the figures would still benefit from error bars or some other indication of the overall statistical error in the findings. I agree that the main contribution of this paper is theoretical, but since the experiments test the empirical validity of the theory, I believe it is nevertheless important to get a sense for the overall deviation in these findings (e.g. across model seeds). If the authors are concerned about a lack of clarity, they could leave the bars out of the main figures but add supplementary figures with error bars. Moreover, some of the lines in Fig. 1 do contain error bars and it would be good to clarify what these error bars represent.

**Questions:**

1. Do you think my suggestion in point 1 of the weaknesses make sense or do you have a reason why you see it as unnecessary?

2. As far as I understand, the reason for the asymmetry between $\nu_g$ and $\nu_h$ in Fig. 2 is the different dimensionality, correct? It would be good to mention these dimensionalities, as I was only able to find them in the appendix.

3. Could you clarify why in Fig. 2, you're using the scalings from Prop 3 rather than from Thm. 5?

**Limitations:**

The authors adequately discuss the limitations of this work.

---

### Official Review · Reviewer_GvvC · 2024-07-13

**Soundness:** 3
**Presentation:** 2
**Contribution:** 3
**Rating:** 6
**Confidence:** 3

**Summary:**

The authors introduce accordion networks (AccNets), which are compositions of multiple shallow networks. By leveraging prior workthat computes norm-based generalization bounds for shallow two-layer networks, the authors bound the complexity of a deep AccNet (as measured by its F1 norm) but the sum of the complexities of the individual shallow networks. They empirically observe that the rates predicted on real-world data are roughly representative of the trained networks, and are indeed much better than those for kernels trained on the same tasks. They put forth a nontrivial scaling law for the excess risk: $N^{-\mathrm{min}(1/2, \nu_g/d_{in}, \nu_h/d_{mid})}$ for an Acc Net compared to $\mathcal L \sim N^{-\mathrm{min}(1/2, \nu_g/d_{in}, \nu_h/d_{in})}$ for a kernel in terms of the dimensionalities $d$ and Sobolev constants $\nu$ of the respective spaces and functions. From this, the authors obtain predictions of several phases, that they put forth experiments to verify.

**Strengths:**

The paper tackles a very important open question in the theory of deep learning, for which not much progress has been made. By creatively leveraging results for shallow network in composition, the authors arrive at a nontrivial bound for deep nets. The empirics are a very compelling and welcome part of the paper. The phase diagrams illustrate the nontrivial predictivity of the theory, especially at the level of the rates. This may have important implications for scaling laws. Modulo minor revisions in discussion and exposition, the whole paper is quite readable for a relatively broad audience.

**Weaknesses:**

I am not sure how compelling the phase plots in Figure 2 are. The bounds in general are extremely loose, however the comparison of the rates in Figure 2c and Figure 3 is very promising. In general, however, it is the experience of the reviewer that measuring a rate is an extremely finicky business. It is therefore important to add a section in the appendix explicitly stating how the rates were obtained and measured. I also strongly encourage the authors to make the code for all figures public.

Because they are used very early on throughout the paper, it is the opinion of the reviewer that the notions of F1 distance and Sobolev norm should be defined earlier on in the paper. Without this, it seems like the audience will be constrained to the set of learning theorists familiar with these terms. However, if these terms are defined early on, the paper becomes remarkably accessible to a much broader audience.

**Questions:**

The plot labels in Figures 2 and 3 are very difficult to read.

A small comment: I have not seen the term "modulo space" used before. Often the term is "quotient space"

The sentence defining the $F_1$ ball (above theorem 1) is confusing, circular, and difficult to read. Please rewrite it.

The excess rate formula $\mathcal L \sim N^{-\mathrm{min}(1/2, \nu_g/d_{in}, \nu_h/d_{mid})}$ is a very important result and I recommend that it be formatted for display, not inline.

How are you measuring "dimension" in 4.1.1? A high-dimensional gaussian with spectral decay of its covariance going as $k^{-\alpha}$ for capacity exponent $\alpha$ is nominally "full dimensional" since it is not strictly speaking constrained to a sub-manifold, and yet basic results in kernel theory and high-dimensional linear regression can show that the generalization error achieves a much better rate at larger values of $\alpha$. Specifically, a model with capacity exponent $\alpha$ and source exponent $r$ achieves a rate of $N^{-2\alpha min(r, 1)}$. See, e.g. https://arxiv.org/abs/2105.15004 .  Such power law anisotropy is in abundant in natural data. In particular shallow two layer networks in the lazy limit can achieve this scaling for such 'easy tasks' with quick spectral decay. On the other hand, the bounds that you state cannot decay faster than $N^{-1/2}$.
* In this sense, it seems that the bounds (shallow or deep) presented are certainly not tight for some datasets. Am I incorrect in concluding this? Do you have an intuition for what causes the breakdown in correctly predicting the error rates in this case?
* Given that they breakdown in that setting, what about the datasets that you study makes it so that the scaling law predictions seem to hold?

**Limitations:**

Given the theoretical nature of this work, it is unlikely to have major social implications.

---

### Comment · Area_Chair_SCc3 · 2024-08-04
**To Authors and Reviewers**

Dear Authors and Reviewers,

Thanks for submitting or reviewing this paper.

Unfortunately, this paper includes a non-anonymous link to their code on Github in footnote 4 on page 13 (in the appendix). This violates the double-blind reviewing policy. Hence this paper will be rejected.

On 13 Jul, reviewers were told that: "It's up to you to decide whether to keep your comments and participate in the rebuttal phase (this does not affect the fact that this paper will be rejected). I suggest keeping the comments as they are still contributing to the authors and the community at large."

I appreciate the reviewers for keeping their comments and I suggest the authors can respond to the questions raised by the reviewers.

Hope all of us can learn something from the discussion.

---

> ### Author Response · Authors · 2024-08-06
>
> We apologize for this mistake. But the reviews are not wasted, they are very useful to us to improve our future submission.

---

### Decision · Program_Chairs · 2024-09-25

**Decision:**

Reject

**Comment:**

This paper includes a non-anonymous link to their code on Github in footnote 4 on page 13 (in the appendix). This violates the double-blind reviewing policy. Hence this paper is rejected.